# Sirtuin-1 sensitive lysine-136 acetylation drives phase separation and pathological aggregation of TDP-43

Jorge Garcia Morato [1,2], Friederike Hans[1,2], Felix von Zweydorf[2], Regina Feederle [3,4], Simon J. Elsässer [5], Angelos A. Skodras[6], Christian Johannes Gloeckner [2,7], Emanuele Buratti[8], Manuela Neumann[2,9] & Philipp J. Kahle [1,2,10 ✉]

Trans-activation response DNA-binding protein of 43 kDa (TDP-43) regulates RNA processing and forms neuropathological aggregates in patients with amyotrophic lateral sclerosis and frontotemporal lobar degeneration. Investigating TDP-43 post-translational modifications, we discovered that K84 acetylation reduced nuclear import whereas K136 acetylation impaired RNA binding and splicing capabilities of TDP-43. Such failure of RNA interaction triggered TDP-43 phase separation mediated by the C-terminal low complexity domain, leading to the formation of insoluble aggregates with pathologically phosphorylated and ubiquitinated TDP-43. Introduction of acetyl-lysine at the identified sites via amber suppression confirmed the results from site-directed mutagenesis. K84-acetylated TDP-43 showed cytoplasmic mislocalization, and the aggregation propensity of K136-acetylated TDP-43 was confirmed. We generated antibodies selective for TDP-43 acetylated at these lysines, and found that sirtuin-1 can potently deacetylate K136-acetylated TDP-43 and reduce its aggregation propensity. Thus, distinct lysine acetylations modulate nuclear import, RNA binding and phase separation of TDP-43, suggesting regulatory mechanisms for TDP-43 pathogenesis.

[1] Laboratory of Functional Neurogenetics, Department of Neurodegeneration, Hertie Institute for Clinical Brain Research, University of Tübingen, 72076 Tübingen, Germany. [2] German Center for Neurodegenerative Diseases (DZNE), 72076 Tübingen, Germany. [3] Institute for Diabetes and Obesity, Monoclonal Antibody Core Facility, Helmholtz Munich, 85764 Neuherberg, Germany. [4] German Center for Neurodegenerative Diseases (DZNE), 81377 München, Germany. [5] Science for Life Laboratory, Department of Medical Biochemistry and Biophysics, Karolinska Institutet, 17165 Stockholm, Sweden. [6] Molecular Imaging Unit, Department of Cellular Neurology, Hertie Institute for Clinical Brain Research, University of Tübingen, 72076 Tübingen, Germany. [7] Core Facility for Medical Bioanalytics (CFMB), Institute for Ophthalmic Research, Center for Ophthalmology, University of Tübingen, 72076 Tübingen, Germany. [8] Molecular Pathology Group, International Centre for Genetic Engineering and Biotechnology (ICGEB), 34149 Trieste, Italy. [9] Department of Neuropathology, University Hospital Tübingen, 72076 Tübingen, Germany. [10] Department of Biochemistry, University of Tübingen, 72076 Tübingen, Germany. ✉email: philipp.kahle@uni-tuebingen.de

The trans-activation response DNA-binding protein of 43 kDa (TDP-43) regulates various RNA-processing steps[1–3] and is found in the neuropathological lesions of patients with amyotrophic lateral sclerosis (ALS) and fronto-temporal lobar degeneration (FTLD)[4,5]. The TDP-43 polypeptide contains two RNA recognition motifs (RRM1 and RRM2), a nuclear localization signal (NLS) and a C-terminal glycine-rich low complexity domain (see Fig. 1a). The NLS allows active nuclear import[6], but TDP-43 can also shuttle between the nucleus and cytoplasm[7,8]. Through its RRMs TDP-43 binds to UG-rich regions of RNA[9,10] and it is involved in RNA transport, stability, and splicing, for example exon 9 of the cystic fibrosis trans-membrane conductance regulator (CFTR) mRNA[11]. The C-terminal low complexity domain of TDP-43 promotes liquid-liquid phase separation (LLPS) as well as pathological protein aggregation[12]. Post-translational modifications (PTMs) can modulate TDP-43 functions in health and disease. Putative pathological PTMs of TDP-43 include C-terminal fragmentations and phosphorylations[13], including the widely used marker pS409/410[14,15]. In addition, lysine modifications have been reported, including ubiquitinations at various residues[16] and sumoylation and acetylation of the RRM1[17,18]. The putative acetyl-mimic [K145Q]TDP-43 was distributed in a stippled manner in trans-fected cells, eventually recapitulating pathological phosphoryla-tion and recruitment of ALS-related factors into TDP-43 aggregates[19]. In addition, [K145Q]TDP-43 had reduced CFTR splicing activity. Thus, lysine modifications may be important for physiology and pathological aggregation of TDP-43. However, the genesis of TDP-43 aggregates remains elusive.

Research on LLPS is unveiling the molecular processes regulating the arrangement of membrane-less organelles. Heteronuclear ribo-nucleoprotein (hnRNP) LLPS is amply reported but whether and how this process is regulated in cells is poorly understood[20]. In the case of TDP-43, RNA binding prevents its phase separation in vitro and reduces droplet formation in vivo[21]. Wild-type (wt)TDP-43 is capable of going into liquid droplets but pathological mutants seem to form more resilient droplets. While the C-terminal glycine-rich domain of TDP-43 seems to play a crucial role in the formation of phase-separated droplets, the N-terminal domain may also contribute to the aggregation process[22–24]. Thus, the regulation of TDP-43 RNA binding could be pivotal in the pathophysiology of TDP-43. The molecular pathways regulating the disengagement from RNA are only beginning to be described.

In the present study, we discovered that acetylation of K84 within the NLS reduced nuclear import whereas acetylation of K136 in the RNA recognition domain impaired TDP-43 RNA binding and splicing capabilities. Such failure of RNA interaction triggered TDP-43 phase separation mediated by the C-terminal low complexity domain, leading to the formation of insoluble aggregates with pathologically phosphorylated and ubiquitinated TDP-43. To con-firm the results from site-directed mutagenesis, we expanded the genetic code via amber suppression to introduce authentic acetyl-lysine at the identified sites. Indeed, [acK84]TDP-43 showed cyto-plasmic mislocalization, and the increased aggregation tendency for [acK136]TDP-43 was confirmed. We generated antibodies selective for TDP-43 acetylated at K84 and K136. Using these tools, we found that the nuclear sirtuin SIRT1 can potently deacetylate [acK136] TDP-43. Moreover, SIRT1 reduced the aggregation propensity of [acK136]TDP-43. Thus, distinct lysine acetylations regulate nuclear import, RNA binding and phase separation of TDP-43 with possible relevance for TDP-43 pathogenesis.

## Results

**Detection of TDP-43 lysine acetylations.** After investigating the sites of TDP-43 lysine ubiquitinations[16], we became interested in how other PTMs could affect TDP-43 functionality and aggregation. Recently acetylation has been linked with TDP-43 pathology[17,19], and lysine acetylation has the potential to disrupt ubiquitination by competing for the same residue. To explore this hypothesis more in depth, we re-examined the mass spectrometry data[16] specifically for acetylated lysines. We found in transfected HEK293E cells that 6xHis-tagged wtTDP-43 was acetylated at lysine residues K79 and K84 around the NLS and K121 and K136 in RRM1 (Fig. 1a). We could not detect the previously published acetylations at K145 and K192[17], perhaps due to slight differences in experimental conditions (see "Discussion").

**Site-directed mutagenesis of TDP-43 acetyl-lysine sites.** Each of the identified acetylated lysine residues was mutagenized to either arginine or glutamine to simulate the lack or the presence of acet-ylation, respectively. We first looked at the subcellular distribution of these N-terminally 6xHis-tagged TDP-43 mutants in transiently transfected HEK293E cells. Immunofluorescence (IF) staining of the 6xHis tag did not show visible deviation from the wtTDP-43 nuclear staining pattern for both acetyl-dead and acetyl-mimic mutations at residues K79 and K121 (Supplementary Fig. 1a). Likewise, the con-servative [K84R]TDP-43 mutant remained mainly nuclear, but a substantial portion of acetyl-mimic [K84Q]TDP-43 was retained in the cytoplasm (Fig. 1b). Quantification showed a significant increase in the percentage of cells with cytoplasmically mislocalized [K84Q] TDP-43 (Fig. 1c). In addition, we performed two different bio-chemical nucleocytoplasmic fractionation assays to support these results. There was a significant increase of [K84Q]TDP-43 in the cytoplasmic fraction when compared to wtTDP-43 (Supplementary Fig. 1b–d). Nevertheless, we could still observe a sizable amount of [K84Q]TDP-43 in the nuclear fraction, as for the [ΔNLS]TDP-43 triple mutant. Thus, alterations at the NLS reduced but did not abolish TDP-43 nuclear import, accounting for the residual splicing activity in the nucleus (see Fig. 3e, f). By contrast, the acetyl-mimic K79Q substitution did not show evident changes in nucleocyto-plasmic distribution of TDP-43, likely because K79 is localized just outside the NLS, whereas K84 is at the core of the NLS[6,16].

The RRM1 mutants K136R and K136Q were predominantly localized in the nucleus, but a significant number of cells displayed a nuclear droplet-like pattern (Fig. 1b, c). This droplet-like nuclear distribution was reminiscent of the RNA-binding deficient F147L/F149L mutant TDP-43[7,25]. For comparison, we generated the previously described[17,19] acetyl-mimic K145Q mutant in RRM1, which showed the expected stippled distribu-tion (Supplementary Fig. 1a). As K136 is in direct contact with bound nucleic acids[26,27], we assume that modifications at K136 disrupt nucleic acid binding and therefore disengage TDP-43 from hnRNP complex localizations. Such dissociated K136-modified TDP-43 may be free to phase separate and self-aggregate into inclusions.

**K136 mutant TDP-43 is pathologically ubiquitinated, phos-phorylated, and insoluble.** We examined if the K136 mutant TDP-43 inclusions in cell culture also showed pathological fea-tures described for human patients[4,5,14,28]. 6xHis-tagged TDP-43 variants were transiently transfected into HEK293E cells with stable knockdown of endogenous TDP-43 (sh^TDP-43)[29] to mini-mize interference from the endogenous wtTDP-43. To assess protein solubility, RIPA-urea solubility fractionation assays were performed. While most of the 6xHis-tagged wtTDP-43 was RIPA soluble, a larger portion of both K136R and K136Q TDP-43 mutants shifted into the RIPA-insoluble fraction (Fig. 2a). Although [F147L/F149L]TDP-43 formed similar intranuclear patches, this designed RNA-binding deficient mutant was as soluble as endogenous TDP-43 (Fig. 2a). It should be noted that

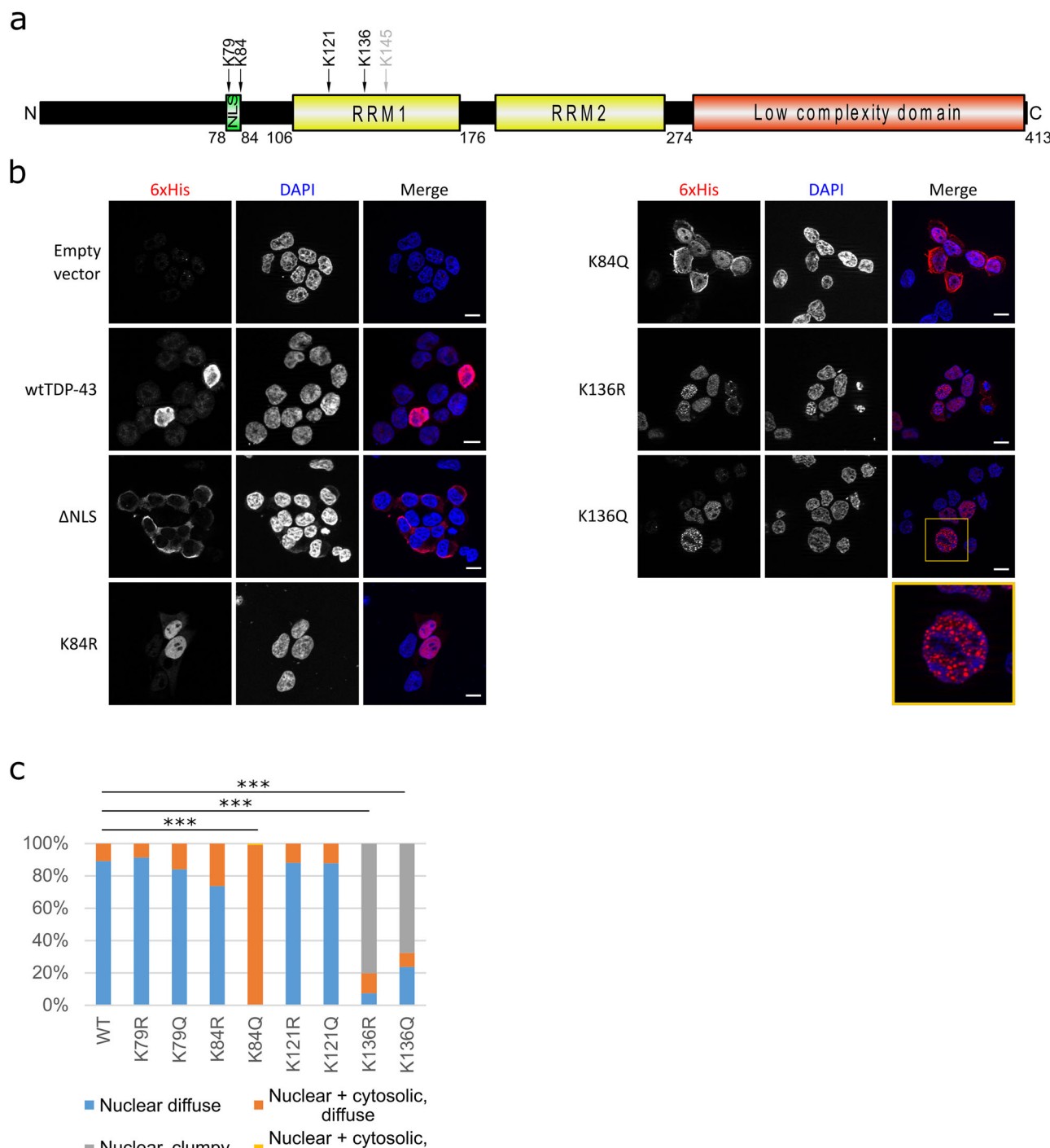

**Fig. 1 TDP-43 is acetylated and its acetylation at K84 may affect its nuclear-cytoplasmic trafficking while acetylation at K136 causes a nuclear droplet-like distribution. a** HEK293E cells were transfected with 6xHis-tagged TDP-43, which was purified with NiNTA beads and subjected to MS. The residues marked in the domain structure of TDP-43 were found to be acetylated (black arrows). K145 has been reported to be acetylated in the literature but we could not find it in our study (grey arrow). **b** Immunostaining of HEK293E cells transfected with 6xHis-tagged wtTDP-43, acetyl-dead (K84R, K136R and K145R) or acetyl-mimics (K84Q, K136Q and K145Q). Scale bar represents 10 μm. **c** Quantification of percentage of cells with 6xHis-TDP-43 protein distribution diffuse nuclear (blue bars), nuclear clumpy (grey bars), nuclear + cytosolic, diffuse (orange bars) and nuclear + cytosolic, clumpy (yellow bars). A total of 300 transfected cells from three independent experiments per group were classified. ***$p < 0.001$ as measured by Chi-squared test. Source data are provided as a Source Data file.

the Flag-tagged [F147L/F149L]TDP-43 was expressed at levels comparable to endogenous TDP-43 in non-silenced cells, whereas the 6xHis-tagged K136 mutants were expressed at higher levels.

To check for pathological PTMs, 6xHis-tagged TDP-43 variants were transiently transfected into sh$^{TDP-43}$-HEK293E cells and purified using NiNTA beads. Both K136R and K136Q TDP-43 were ubiquitinated more strongly than wtTDP-43 even after proteasomal inhibition (Fig. 2b). In addition, K136R and K136Q mutant TDP-43 showed phosphorylation at the extreme C-terminal serine residues S409/410 detected with a phospho-

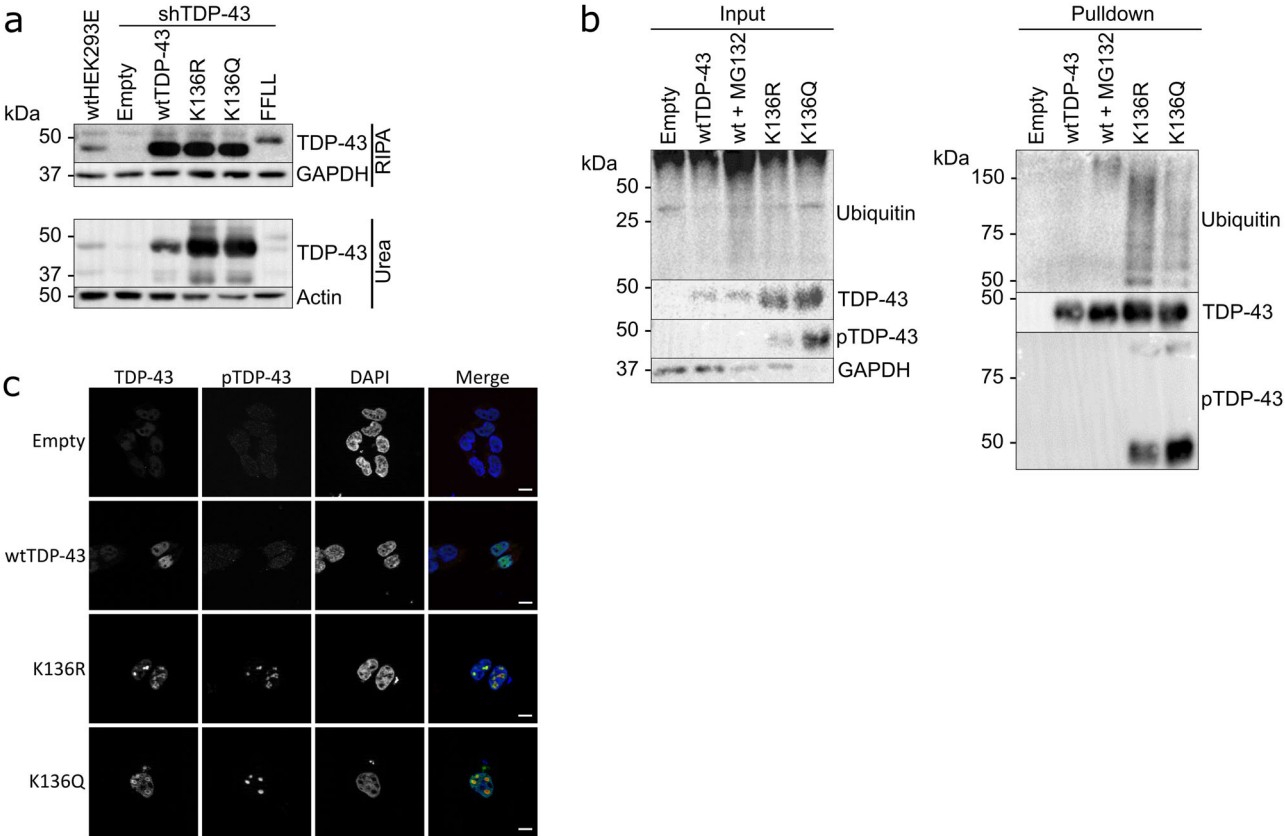

**Fig. 2 K136 mutations reduce the solubility of TDP-43 and trigger its ubiquitination and pathological phosphorylation. a** HEK293E sh[TDP-43] cells were transfected with 6xHis-tagged TDP-43 (wt, K136R and K136Q) or Flag-tagged FFLL TDP-43. RIPA soluble and insoluble fractions were separated and analyzed by Western blotting. $N = 3$ biologically independent samples. Source data are provided as a Source Data file. **b** Sh[TDP-43]-HEK293E cells were transfected with 6xHis-tagged TDP-43 (wt, K136R and K136Q). Proteasome activity was inhibited by treatment with 20 μM MG-132 for 6 h, which caused wtTDP-43 ubiquitination (lane 3). 6xHis-tagged TDP-43 was purified with NiNTA beads and analyzed on Western blots. $N = 3$ biologically independent samples. Source data are provided as a Source Data file. **c** Transfected sh[TDP-43]-HEK293E cells were double immunostained for TDP-43 and phosphorylated TDP-43 (S409/410). Scale bar represents 10 μm. $N = 3$ biological replicates.

specific antibody[15] (Fig. 2b). This pathological phosphorylation was also visible at the nuclear inclusions of K136 mutant TDP-43 by immunostaining (Fig. 2c). Thus, the nuclear inclusions formed by K136 mutant TDP-43 recapitulate PTMs found in patients.

**K136 mutant TDP-43 fails to bind RNA and lacks splicing activity.** Because K136 in the RRM1 is important for interaction with nucleic acids[26,27], we hypothesized that alterations at position K136 could interfere with RNA binding of TDP-43 and alter its splicing capabilities. First, we performed a RNA-protein pulldown assay of TDP-43 with its preferred binding sequence poly-$(UG)_{12}$[9]. While wtTDP-43 strongly bound to synthetic poly-$(UG)_{12}$ but not the negative control poly-$(UC)_{12}$ RNA, the acetyl-mimic [K136Q]TDP-43 showed a reduction in RNA binding (Fig. 3a). Additionally, we measured the RNA-binding strength using a two-filter trap assay using an excess of synthetic poly-$(UG)_{12}$ as target. The first nitrocellulose membrane will capture the biotinylated RNA probe only when bound to protein and the second nylon membrane captures the unbound flow-through RNA. This assay clearly showed the concentration-dependent formation of [wt]TDP-43 protein complex with poly-$(UG)_{12}$ RNA retained on the nitrocellulose membrane (Fig. 3b). In contrast, the acetyl-mimic [K136Q]TDP-43 had significantly reduced poly-$(UG)_{12}$ RNA-binding strength (Fig. 2b–d).

To assess the splicing activity of TDP-43 in cells, we used the established *CFTR* minigene reporter assay[11]. To minimize the splicing activity of the endogenous wtTDP-43, we used sh[TDP-43]-

HEK293E cells[29]. Mutant constructs of TDP-43 were cotransfected together with a plasmid containing exons 9-11 of *CFTR* in a minigene. In parental, non-silenced cells exon 9 was skipped, which was severely blunted in sh[TDP-43] cells (Fig. 3e). Ex9 splicing could be rescued by re-transfecting [wt]TDP-43 (Fig. 3e). The nuclear [K84R]TDP-43 rescued *CFTR* splicing to wt levels. [K84Q]TDP-43 rescued *CFTR* splicing activity to a lesser extent than wt or [K84R]TDP-43 (Fig. 2h). The reduced splice activity of the nuclear import impaired [K84Q]TDP-43 was similar to that of [ΔNLS]TDP-43, suggesting that the residual nuclear TDP-43 (Fig. 1d) was sufficient to promote ex9 skipping. Indeed, in our experience, very strong reduction of TDP-43 activity is necessary to cause loss of TDP-43 splice activity in cells. In contrast, both K136R and K136Q mutants showed significantly reduced *CFTR* exon 9 splicing (Fig. 3e, f), linking the formation of nuclear inclusions with TDP-43 loss of function. The previously described[17,19] K145Q mutant TDP-43 also showed some reduction of *CFTR* splice activity, although the effect was less pronounced than for the K136 mutants expressed at comparably high protein levels (Fig. 3e). Together, both *CFTR* splicing assay and RNA-protein pulldown show that alterations at position 136 severely perturb the RNA-binding and splicing capabilities of TDP-43.

It was puzzling that both the acetyl-mimic K136Q and the acetyl-dead K136R substitutions showed the same aggregation-promoting effects. The K136 residue may be in a structurally restrained position not tolerating even conservative amino acid

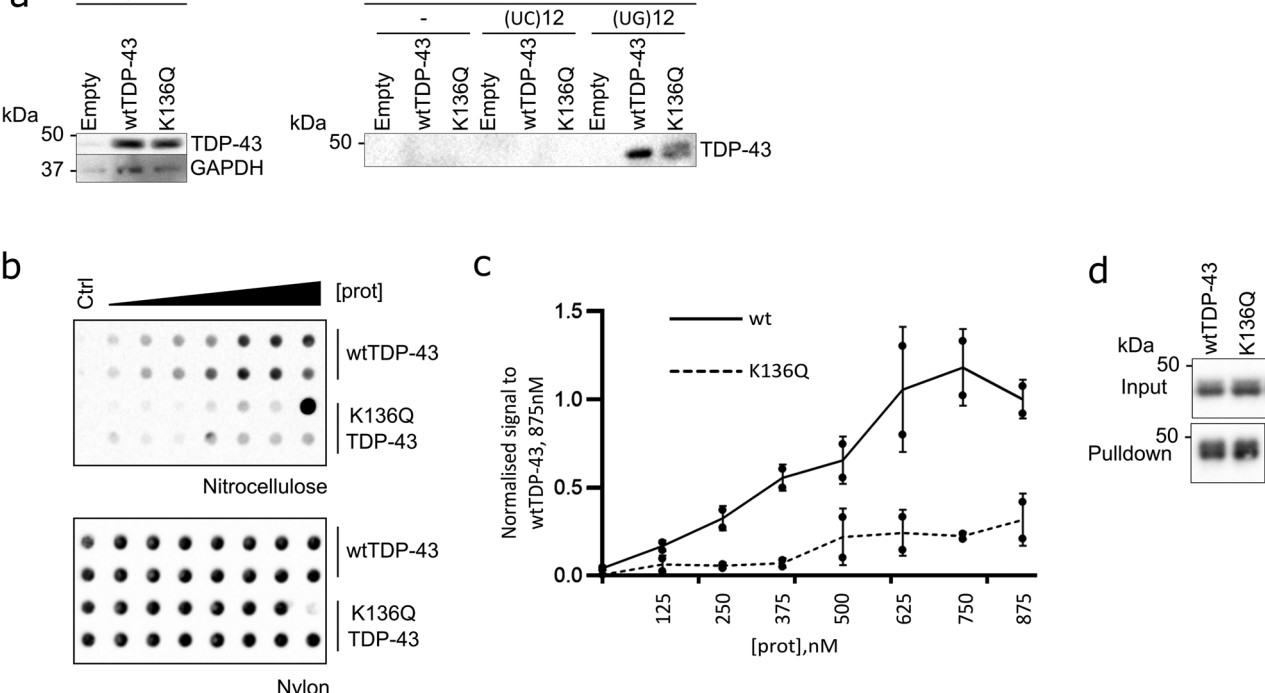

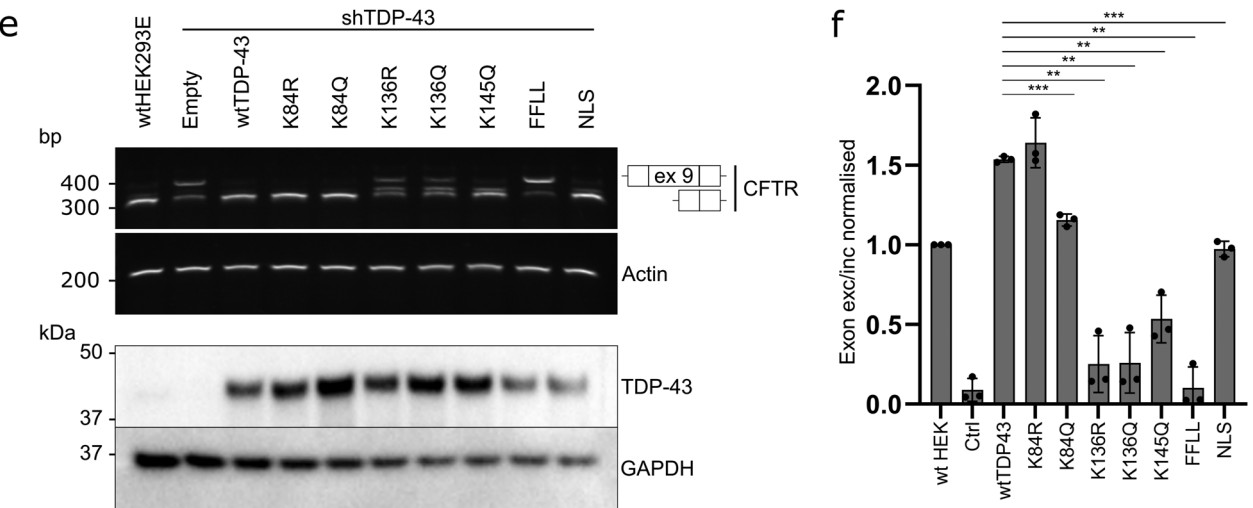

**Fig. 3 K136 mutations alter the RNA binding and splicing capabilities of TDP-43. a** HEK293E sh^TDP-43 cells were transfected with an empty plasmid, wt or K136Q TDP-43. The lysates were then incubated with biotinylated $(UC)_{12}$ or $(UG)_{12}$ RNA oligomers. Protein-RNA mix was incubated with magnetic streptavidin beads. Bound proteins were eluted and analyzed by Western blotting. $N = 3$ biological replicates. Source data are provided as a Source Data file. **b** Increasing concentrations of 6xHis purified TDP-43 proteins were incubated with biotinylated $(UG)_{12}$ oligomers. The resulting complexes were analyzed by a filter-binding assay. Membranes were incubated with HRP-coupled streptavidin. One of the [K136Q]TDP-43 data points illustrates overload problems at high protein concentration (>875 nM), precluding firm establishment of saturation binding curves. Source data are provided as a Source Data file. **c** Quantification of three replicates of filter-binding assay of wt (full line) and K136Q (dashed line) TDP-43 to biotinylated $(UG)_{12}$ RNA oligomers. **p < 0.01. Data are presented as mean values ± SD. Unpaired, two-sided *t*-test was used for comparison. Source data are provided as a Source Data file. **d** Representative Western blot of the protein levels before and after the purification of native TDP-43 via NiNTA pulldown. $N = 3$ biological replicates. Source data are provided as a Source Data file. **e** HEK293E sh^TDP-43 cells were cotransfected with the specified TDP-43 constructs and a plasmid containing the *CFTR* minigene. RNA was extracted and splicing of *CFTR* exon 9 was assessed via rtPCR (upper panel). Protein levels are shown in the lower panel. $N = 3$ biologically independent samples. **f** Quantification of three replicates assessing effects of mutations at different lysines of TDP-43 on the splicing of *CFTR* exon 9. Data are presented as mean values ± SD. Unpaired, two-sided *t*-test was used for comparison. **p < 0.01. Source data are provided as a Source Data file.

substitutions. Indeed, the NMR structure of the TDP-43 RRMs complexed with RNA[26] revealed that the side chain of K136 is in close apposition to the nucleic acid and the key RNA-binding residues F147/F149 (Supplementary Fig. 2a). Moreover, K136 is within ≈3 Å distance to Q134, L139 and T199, evidently forming a tight RNA-binding configuration that appears not to accommodate the larger guanidinium group of the otherwise conservative K136R substitution. Supplementary Fig. 2b highlights the K136 residue within the RRM1 bound to RNA. Modelling the acetyl-dead K136R mutation, which surprisingly caused similar aggregation propensities as the acetyl-mimicking K136Q substitution, revealed clashes with the protein backbone. Thus, classical site-directed mutagenesis might introduce local perturbations to the RRM1 structure, which may not entirely reflect the effects of authentic lysine acetylation. Indeed, the model for [acK136]TDP-43 showed hardly any clashes with the protein backbone, but a collision with the bound RNA structure. In order to distinguish mutagenesis artefacts from authentic lysine acetylation effects, we needed to introduce acK at position 136 of TDP-43.

**Introduction of acetyl-lysine (acK) through amber suppression.** As K136 turned out to be a very delicate residue for site-directed mutagenesis, it was important to properly validate the pro-aggregative effects for [acK136]TDP-43. We succeeded to incorporate authentic acK at the apparently regulatory sites in TDP-43 with amber suppression methodology. TDP-43 mutants were generated with a C-terminal 6xHis tag and an amber stop codon (TAG) substituting the codon at position 84 and at position 136, respectively. Each of these plasmids was cotransfected into sh$^{TDP-43}$-HEK293E cells, together with a plasmid containing a N-terminally Flag-tagged chimeric acetylated lysine RNA synthetase (AcRS) and four copies of tRNA$_{TAG}$[30]. Together the designed AcRS and the tRNA$_{TAG}$ will incorporate acK added to the media at the introduced amber stop codon, directly incorporating an acetylated lysine site specifically during the protein translation.

In absence of acK, only truncated forms of TDP-43 were visible due to the introduction of the amber stop codons (Fig. 4a). After adding 5 mM acK in the media for 24 h, the constructs were translated beyond the amber sites such that full-length TDP-43 including the C-terminal 6xHis tag became detectable, confirming successful amber suppression (Fig. 4a). There was a reduction in the amount of truncated TDP-43 but it did not completely disappear, pointing out the slight inefficiency of the system when compared to endogenous tRNAs. The expression of amber-suppressed full-length mutant TDP-43 was not as high as that from a wtTDP-43 cDNA (Fig. 4a), but still robust. Importantly, [acK84]TDP-43 showed a similar reduction of nuclear import (Fig. 4b) as the K84Q mutant (Fig. 1b), and [acK136]TDP-43 showed the same spotty nuclear distribution (Fig. 4b) as the K136Q mutant (Fig. 1b). To check if the RNA splicing capabilities of TDP-43 were compromised after introducing an acetylated lysine at position 136 in a similar manner to the K136R and K136Q mutants, we performed the *CFTR* minigene assay in sh$^{TDP-43}$-HEK293E cells. Indeed, [acK136]TDP-43 could only partially rescue *CFTR* exon 9 splicing (Fig. 4c). In addition, we also introduced acetylated lysine at K79 and K121 (Supplementary Fig. 3). As expected from the point mutations (Supplementary Fig. 1a), [acK79]TDP-43 and [acK121]TDP-43 did not show any different distribution than wtTDP-43. Taken together, the acetyl-mimic lysine-to-glutamine substitutions did reflect authentic lysine acetylation properties, as confirmed by amber suppression introduction of acK directly.

We were then interested if [acK136]TDP-43 could form phase-separated droplets or even aggregates in the cytoplasm, the most

common site where human neuropathological TDP-43 is found[31]. To study this, we disrupted the NLS[6] in the construct ΔNLS-K136$_{TAG}$. [AcK136)TDP-43ΔNLS was uniformly distributed throughout the cytoplasm in most cells. However, there was a fraction of cells developing multiple cytoplasmic aggregates (Fig. 4d–f). Thus, when nuclear import is impaired, K136 acetylation may also promote aggregation of cytosolically mislocalized TDP-43. Next, we deleted the C-terminus in the construct ΔC-K136$_{TAG}$ because the C-terminal low complexity domain mediates LLPS and aggregation of TDP-43[32]. [acK136)TDP-43ΔC showed a homogeneous nuclear distribution (Fig. 4d–f), confirming that the C-terminal part of TDP-43 plays a crucial role in phase separation and aggregation.

**FRAP analysis and dynamics of K136 mutant TDP-43 aggregation.** As we confirmed that [K136Q]TDP-43 showed the same properties as amber suppressed [acK136]TDP-43, we characterized in greater detail the aggregation process of a C-terminally EGFP-tagged [K136Q]TDP-43 construct. First we measured protein motility in the droplet-like inclusions by fluorescence recovery after photobleaching assay (FRAP). HEK293E cells were transfected with either wtTDP-43-EGFP or [K136Q]TDP-43-EGFP. Then regions of interest (ROI) with bleached for half a second by 488 nm laser light irradiation and allowed to recover (Fig. 5a). The wtTDP-43-EGFP showed recovery within a minute after photobleaching (Fig. 5b), consistent with the motility of TDP-43 in the nucleus[21]. By contrast, two types of less mobile pools of [K136Q]TDP-43-EGFP were detected (Fig. 5b). Large [K136Q]TDP-43-EGFP aggregates (ø ≥ 1 μm) did not recover for up to 3 min recording (Fig. 5b), likely constituting solid aggregates. The smaller punctae (ø ≤ 0.5 μm) showed a significantly slower recovery than the wtTDP-43 with an immobile fraction not recovering at all (Fig. 5b, c), indicative of LLPS[21].

To investigate the kinetics of LLPS and protein aggregation, we performed long-term live imaging of [K136Q]TDP-43-EGFP. Transfected HEK293E cells were imaged for several hours and the distribution of TDP-43 was monitored. [K136Q]TDP-43 was recorded going into phase separation, forming droplets and these droplets were seen fusing with one another (Fig. 5d and Supplementary Videos 1–3). To examine the [K136Q]TDP-43 aggregate-formation kinetics in detail we quantified the size of aggregates in HEK293E cells fixed 24, 48 and 72 h after transfection. The quantification showed that after 48 h there was an increase in the percentage of TDP-43 aggregates larger than 10 μm, and the size difference became significant after 72 h (Fig. 5e). Together, these results suggest that the aggregation of TDP-43 is a dynamic process from a diffuse distribution to phase separation, eventually towards larger solid aggregates.

**Generation of site-specific antibodies against lysine-acetylated TDP-43.** To study the acetylated forms of TDP-43 in greater detail, we developed rat monoclonal antibodies against acK84 and acK136 TDP-43. Two clones were selected that recognized with high sensitivity their corresponding amber-suppressed TDP-43 targets (Fig. 6a, b) in Western blots, with a very faint band visible in the wtTDP-43 lane. The antibodies did not recognize K84R or K136R TDP-43 mutants, respectively. Both antibodies were also capable of detecting the corresponding lysine acetylated forms of TDP-43 by IF staining (Fig. 6c, d).

**SIRT1 can deacetylate [acK136]TDP-43 and revert its aggregation propensity.** These antibodies allowed us to check for enzymes that could deacetylate [acK136]TDP-43. In humans, there are 18 protein lysine deacetylases[33]. First we tested the histone deacetylases (HDACs) 1–8, representing class I and class II

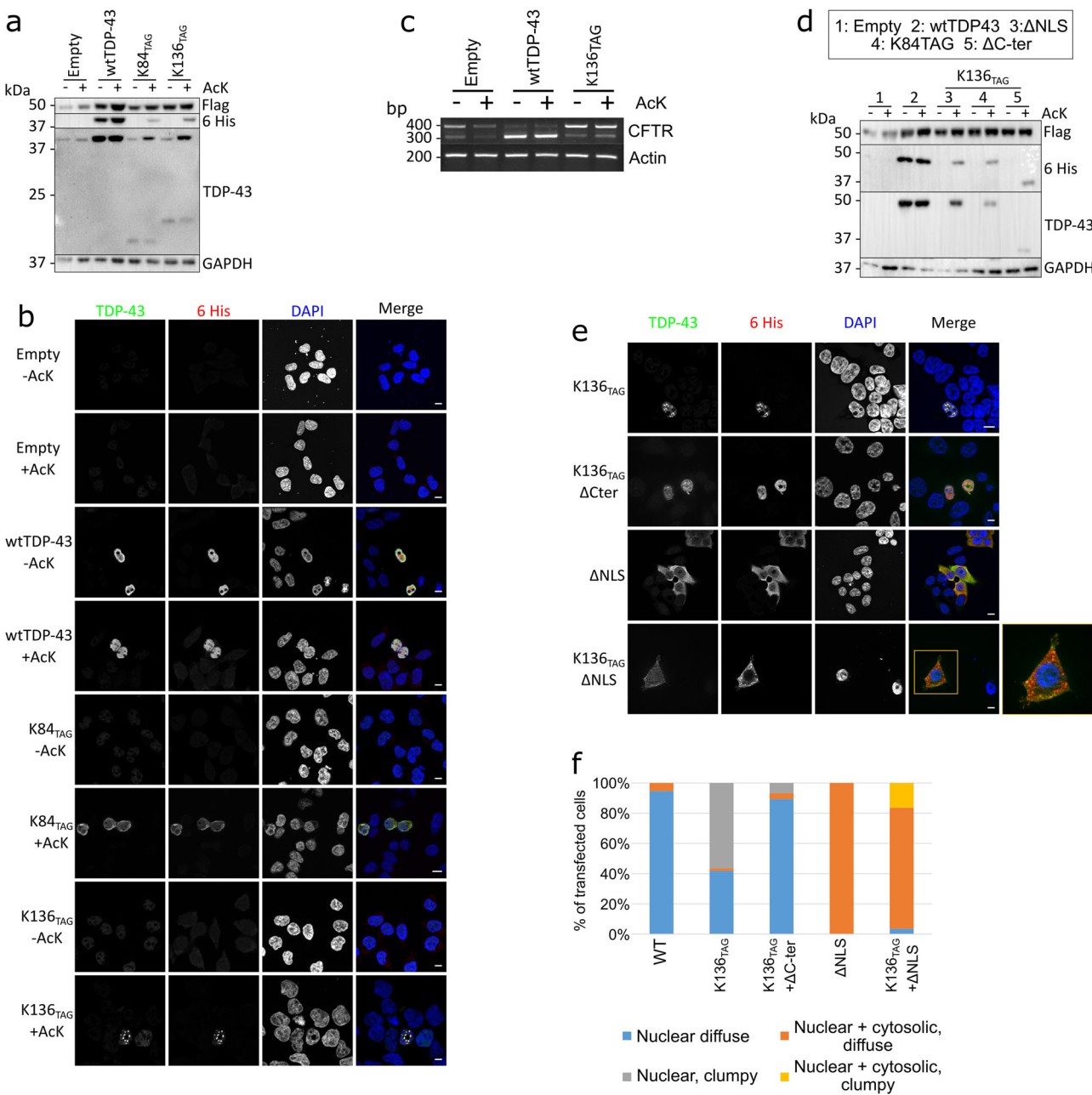

**Fig. 4 Amber suppression-mediated acetylation at K136 of TDP-43 prevents *CFTR* splicing, triggers the formation of nuclear droplets via its C-terminal domain.** Acetylation at K84 reduces nuclear import. **a** HEK293E cells cotransfected with a plasmid containing AcRS and 4xtRNA_TAG (E451) and with TDP-43 with an amber stop codon at different positions and a C-terminal 6xHis tag. Media was changed to 5 mM acK-containing media 24 h before lysis. Protein lysates were analyzed via Western blot. N = 3 biological replicates. Source data are provided as a Source Data file. **b** Cells were cotransfected with the E451 plasmid and different C-terminally 6xHis-tagged TDP-43 constructs containing amber suppression codons. Cells were fixed after 24 h in the presence of 5 mM acK. Scale bar represents 10 μm. N = 3 biological replicates. **c** HEK293E sh^TDP-43 cells cotransfected with E451, *CFTR* minigene, and an empty plasmid, wt or K136_TAG TDP-43. After 24 h in the presence of 5 mM acK, RNA was extracted and the splicing of *CFTR* exon 9 was assessed via rtPCR and visualized in an agarose gel. N = 3 biological replicates. Source data are provided as a Source Data file. **d** HEK293E sh^TDP-43 cells were cotransfected with E451 plasmid and different C-terminally tagged TDP-43 constructs. Lanes 3–5 have an amber stop codon at position 136 in addition to a mutated NLS, an amber stop codon at position 84 or a deletion of the C-terminus. Cells were treated with 5 mM acK for 24 h before lysis. N = 3 biological replicates. **e** HEK293E sh^TDP-43 cells were cotransfected with E451 and different C-terminally tagged TDP-43 constructs. Cells were fixed and immunolabeled after 24 h in the presence of 5 mM acK. Scale bar represents 10 μm. N = 3 biological replicates. **f** Quantification of percentage of cells with 6xHis-TDP-43 protein distribution diffuse nuclear (blue bars), nuclear clumpy (grey bars), nuclear + cytosolic, diffuse (orange bars) and nuclear + cytosolic, clumpy (yellow bars). A total of 250 cells per group cotransfected with E451 plasmid and different 6xHis C-terminally tagged TDP-43 constructs. Source data are provided as a Source Data file.

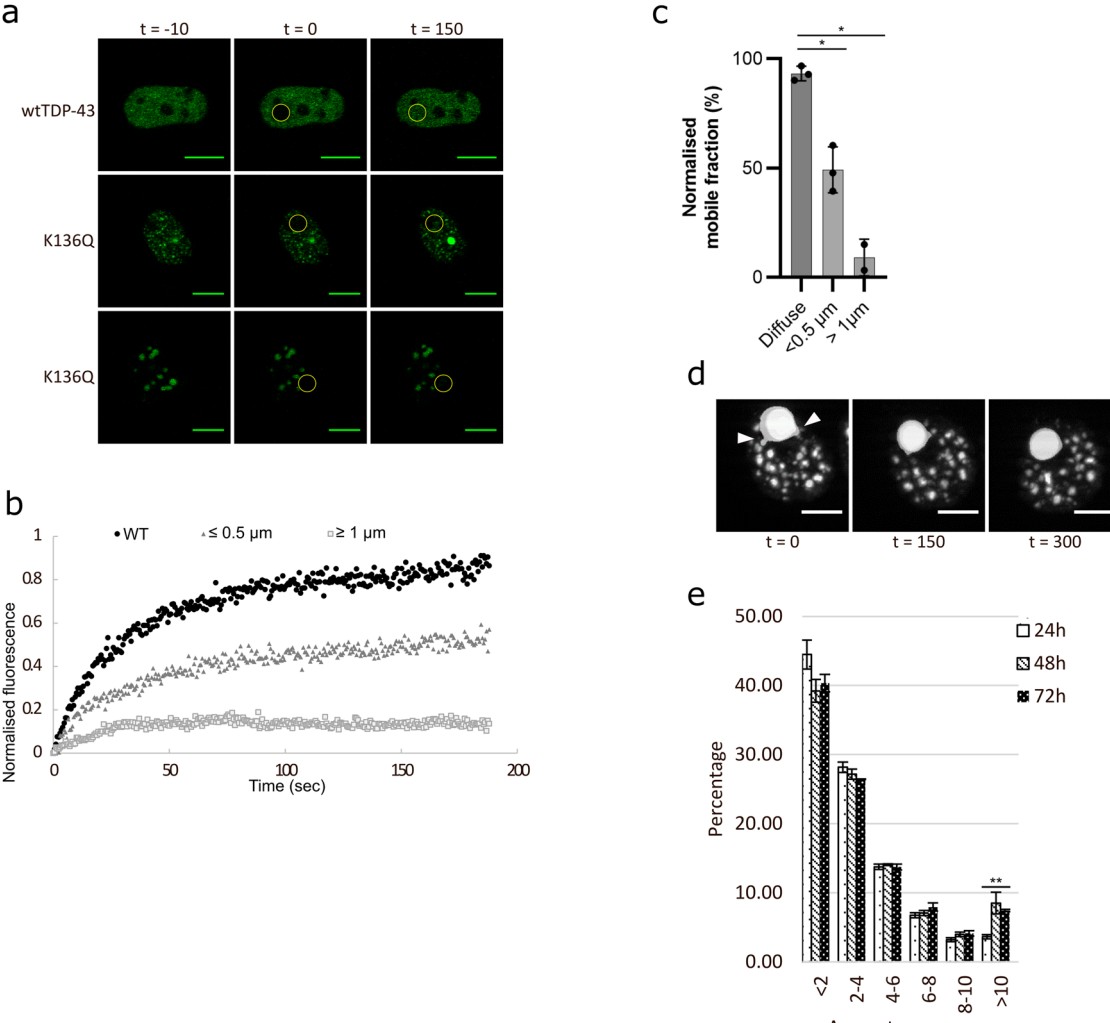

**Fig. 5 [K136Q]TDP-43 droplets have reduced mobility, growing in size and fusing over time. a** Representative images of FRAP analysis of HEK293E cells transfected with C-terminally EGFP-tagged wt and K136Q TDP-43. Two different groups of [K136Q]TDP-43-EGFP droplets based on size were determined. Time in seconds. Scale Bar represents 5 μm. **b** Quantification of FRAP images (three cells quantified per group). The inclusions bodies are classified as follows: diffuse TDP-43 (circles), small inclusions (triangles) and big inclusions (squares). Source data are provided as a Source Data file. **c** Quantification of mobile fraction of FRAP analysis of wt and K136Q TDP-43 mutants. Three cells quantified per group. Data are presented as mean values ± SEM. Unpaired, two-sided t-test was used for comparison *$p < 0.05$. Source data are provided as a Source Data file. **d** Selected frames from green fluorescence live-cell imaging of cells expressing [K136Q]TDP-43-EGFP. Close-up of granules fusing (white arrows). Time in seconds. Scale bar represents 5 μm. $N = 3$ biological replicates. **e** Histogram of K136Q TDP-43 aggregate size in HEK293E fixed 24, 48 or 72 h after transfection. A total of 100 transfected cells analyzed for each time point. Different patterns represent the respective hours after transfection. Unpaired, two-sided t-test was used for comparison. Data are presented as mean values ± SEM. **$p = 0.0075$. Source data are provided as a Source Data file.

deacetylases. Each of these HDACs was cotransfected into amber suppressed [acK136]TDP-43 cells. None of the HDAC1-8 effectively reduced TDP-43 K136 acetylation, (Supplementary Fig. 4a). However, the results for Myc-tagged HDAC1 and HDAC6 were somewhat inconclusive as their expression interfered with the amber suppression efficiency for [acK136]TDP-43 expression (Supplementary Fig. 4b). Next, we tested all class III deacetylases (sirtuins), also because of their involvement in neurodegenerative diseases including ALS[34]. Flag-tagged SIRT1-7 were cotransfected with K136$_{TAG}$ and the deacetylation effects on amber-suppressed [acK136]TDP-43 were probed with the acetylation-specific antibody. Among the 7 sirtuins, only SIRT1 and SIRT2 strongly reduced [acK136]TDP-43 immunoreactivity (Supplementary Fig. 4c). To confirm that the reduction in K136 acetylation was truly caused by the deacetylation activity of these sirtuins, we repeated the experiment in the presence of Ex527, a selective SIRT1 and to a lesser extent SIRT2 inhibitor. Indeed, treatment with Ex527 dose-

dependently reduced the deacetylation of [acK136]TDP-43 by SIRT1 and SIRT2 (Fig. 7a). [AcK136]TDP-43 deacetylation was observed after extreme overexpression of the mainly cytoplasmic NAD+ dependent deacetylase SIRT2[33], so we cannot rule out overexpression artefacts. However, moderate levels of the nuclear deacetylase SIRT1[33] were sufficient to counteract TDP-43 acetylation at K136. It is of note that TDP-43 was reported to bind to the 3′-UTR of SIRT1 mRNA and regulate SIRT1 expression[35]. Thus, the nuclear deacetylase SIRT1 may be engaged in a regulatory network with TDP-43. To see if the reduction of acetylation correlated with a reduction in the amount of aggregates of [acK136]TDP-43, we cotransfected those sirtuins with an impact in TDP-43 acetylation (SIRT1 and SIRT2) as well as other sirtuins with similar level of expression (SIRT3 and SIRT6). When compared to cells without any sirtuin overexpression, SIRT1 and 2 caused a significant reduction in the number of cells with inclusions, even though some level of TDP-43 acetylation was detected (Fig. 7b, c).

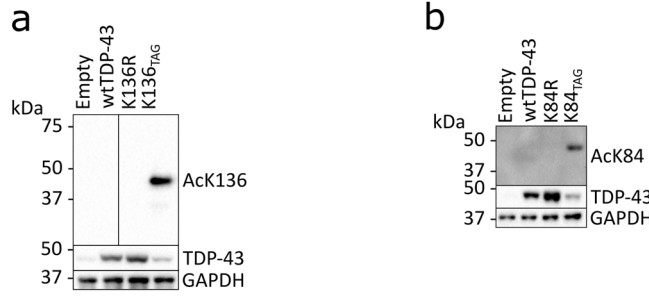

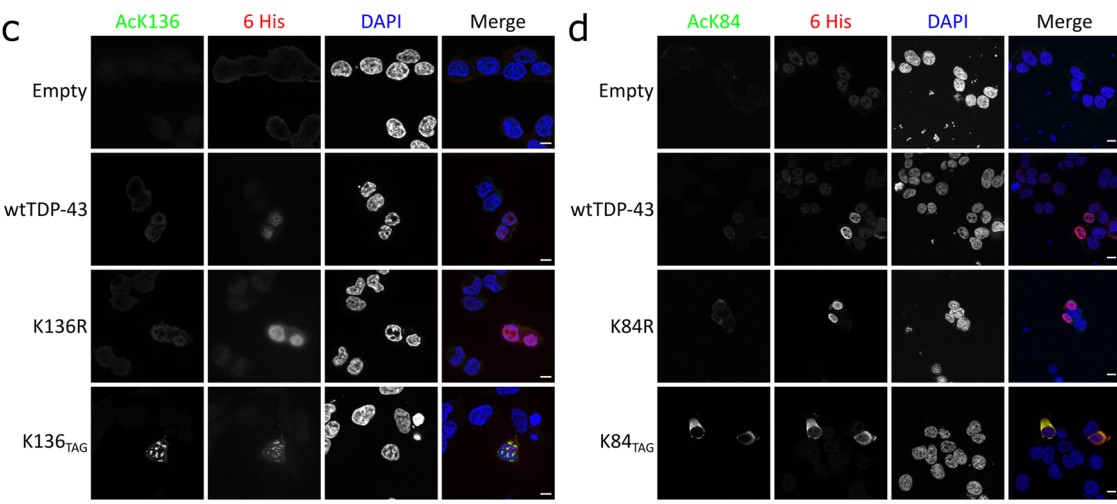

**Fig. 6 Antibodies against acetylated TDP-43 specifically recognize the modified forms. a**, **b** HEK293E sh^TDP-43 cells were co transfected with E451 plasmid and different C-terminally 6xHis-tagged TDP-43 constructs. After 24 h in the presence of 5 mM of acK cells were lysed. Samples were Western probed with antibodies against acetylated K84 (clone 9G9) and K136 (clone 14D4), respectively. N = 3 biological replicates. Source data are provided as a Source Data file. **c**, **d** HEK293E sh^TDP-43 cells were cotransfected with E451 plasmid and different C-terminally 6xHis-tagged TDP-43 constructs. After 24 h in the presence of 5 mM of acK cells were fixed and immunostained for 6xHis or acetylated TDP-43. Scale bar represents 10 μm. N = 3 biological replicates.

## Discussion

In ALS only 4% of familial cases have mutations in TDP-43 and they are even rarer in FTLD (<1%)[36,37]. PTMs of wtTDP-43 might promote disease but it is still unknown if they are a cause or consequence of TDP-43 aggregation[5,17]. Here we show that TDP-43 nuclear import and RNA binding are dynamic processes that can be regulated by distinct PTMs. Specifically, we have discovered that acetylation of the core NLS residue K84 partially reduces nuclear import of TDP-43, leading to the accumulation of cytosolically mislocalized protein, which is considered as an early step of TDP-43 pathogenesis. Moreover, we identify K136 in the RRM1 as a potential regulatory site for TDP-43 RNA binding and splicing activity. Although we could confirm the stippled distribution and reduced *CFTR* splicing activity for the previously reported [K145Q]TDP-43[17,19], the effects for K136 located directly within an essential RRM1 site appeared stronger in our experimental system. Acetylation can modulate the binding to RNA of other hnRNPs and therefore likely for TDP-43 as well[38,39]. Such loss of functional RNA binding may lead to LLPS and eventually the formation of intranuclear TDP-43 aggregates. Acetylation of lysine residues in the RRM1 could trigger aggregation of TDP-43, driving it into phase-separated droplets that eventually coalesce into bigger aggregates. This was also very recently proposed by Yu et al. who looked at [K145Q/K192Q] TDP-43, two acetylation sites that interfere with TDP-43 RNA-processing functions[40]. The phase-separated [K145Q/K192Q] TDP-43 can become less fluid and presumably turn into solid

aggregates. Our findings support this progression for [acK136] TDP-43 and identify SIRT1 as a deacetylase counteracting this process. When combined with ΔNLS, [acK136]TDP-43 also showed a tendency to form cytosolic aggregates, which are much more commonly found in human patients than intranuclear inclusions[41]. Thus, acetylation of K136 might disengage TDP-43 from functional hnRNP complexes, liberating the protein to unmix into liquid phases. Such LLPS transitions may lead to TDP-43 self-aggregation and the formation of pathological inclusions. The observation that SIRT1 can deacetylate [acK136] TDP-43 indicates that this is a dynamic process that can be regulated in cells. The balance of TDP-43 protein lysine acetyl-transferases and deacetylases might determine its physiological functions and when derailed, could lead to LLPS and pathological TDP-43 aggregation in disease. Modulation of this pathway could offer novel therapeutic approaches for the treatment of FTLD and ALS.

Residue K136 was recently discovered as a potential TDP-43 sumoylation site[18]. Consistent with this study by Maurel et al. (2020) using TDP-43 GFP fusion proteins, we found that 6xHis-tagged [K136R]TDP-43 formed intranuclear inclusions. The similarities in behaviour between K136R and K136Q TDP-43 mutants were puzzling, suggesting that instead of mimicking the presence or lack of acetylation, the mutants were introducing backbone changes in TDP-43. Point mutagenesis is a broadly used approach to study the cellular effects of protein lysine acetylation but in some cases can fail to rescue protein function[42].

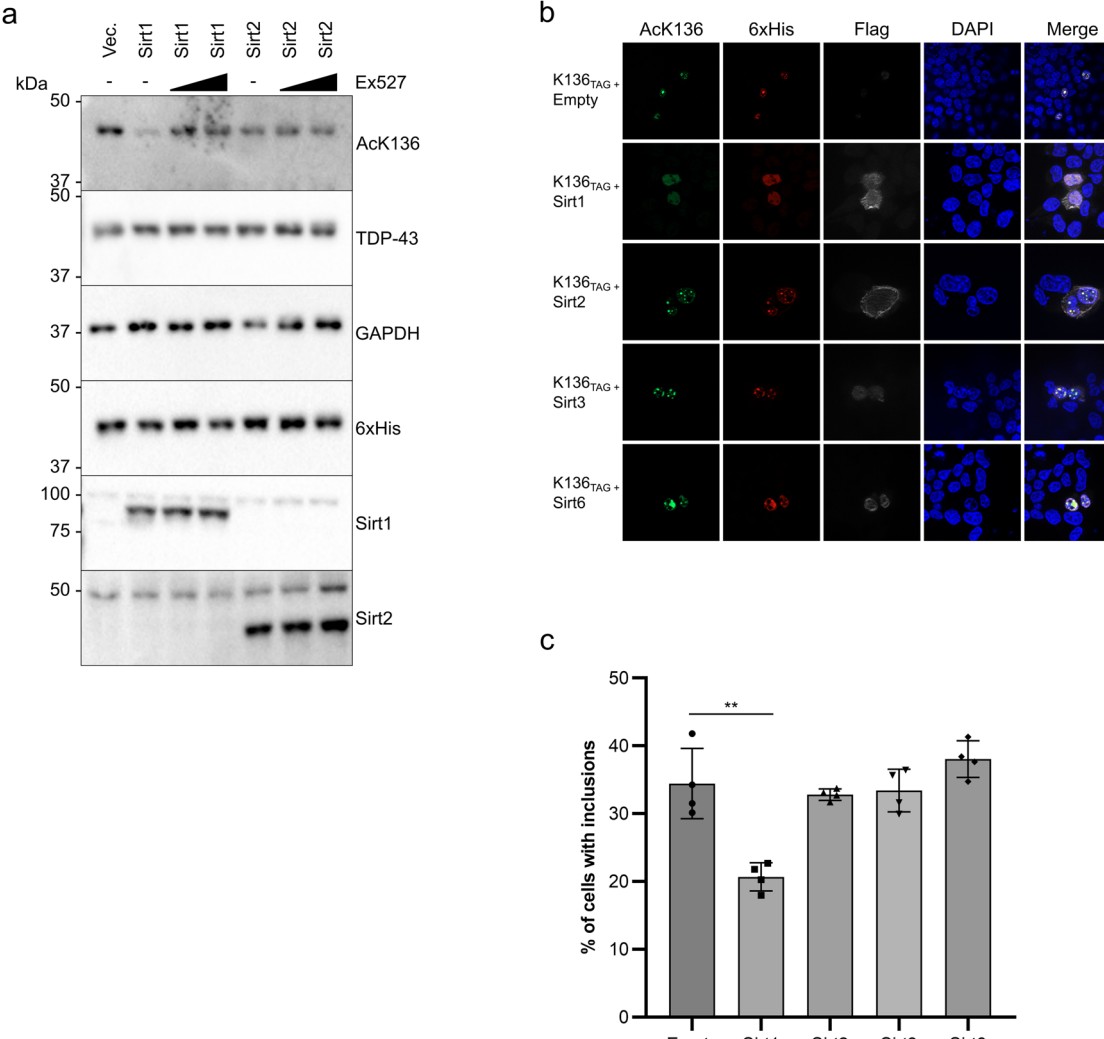

**Fig. 7 Sirtuins 1 and 2 can deacetylate K136 of TDP-43 and prevent its aggregation. a** HEK293E sh[TDP-43] cells were cotransfected with E451, K136[TAG] TDP-43 and SIRT1 or SIRT2. Twenty-four hours before lysis, 5 mM acK and DMSO, 1 µM or 10 µM Ex527 containing media was supplied to the cells. $N = 3$ biological replicates. Source data are provided as a Source Data file. **b** Stably amber suppressed sh[TDP-43]-HEK293E cells were cotransfected with different sirtuin constructs together with K136[TAG] TDP-43. Cells were fixed after 24 h in the presence of 5 mM acK. Scale bar represents 10 µm. **c** Quantification of the number of cells with TDP-43 positive inclusions (6xHis staining). A total of 200 cells from 4 independent experiments were recorded per condition. Unpaired, two-sided $t$-test was used for comparison. Data are presented as mean values ± SEM. **$p = 0.0083$. Source data are provided as a Source Data file.

In addition, glutamine and arginine contain delocalized electrons that are critical for the formation of phase-separated entities[43]. This provides an explanation for the stronger occurrence of phosphorylated aggregates in K136R/Q TDP-43. Amber suppression has its own disadvantages in terms of efficiency and availability of synthetic amino acids, in addition to the possibility of side effects. Nevertheless, transcriptomic analysis of stably amber-suppressed cells did not show upregulation of protein folding stress genes[44]. Another benefit of this approach is the possibility of removing the modification in vivo via deacetylases (see Fig. 7).

We could not detect phosphorylation and ubiquitination of the amber suppressed [acK136]TDP-43, in contrast to the K136Q mutant TDP-43. Three factors might explain this negative finding. First, compared with transfected mutant [K136Q]TDP-43, expression of amber-suppressed TDP-43 is much lower (see Fig. 6a), potentially below detection limit of other PTMs. The lower concentration might also not be sufficient to seed protein aggregation, phosphorylation and ubiquitination. Second, K136-

acetyl in the amber-suppressed TDP-43 can be removed by endogenous deacetylases such as SIRT1. This would attenuate any downstream effects of this acetylation. And third, as shown by Wang et al. glutamines favour the hardening of phase-separated RNA-binding molecules[24]. While these key differences could be behind the lack of the strongest markers of TDP-43, amber suppression still could help identifying K136 as a residue crucial for RNA splicing and LLPS in TDP-43.

In our MS analysis, we did not detect the previously reported acetylations at K145 and K192 in the RNA-binding domain of TDP-43[17]. Although our experimental setups were similar, perhaps the use of the protein aggregation optimized QBI-293 subclone in their studies made the difference, whereas our study in HEK293E cells revealed acetylation of K121 and K136 in RRM1. Moreover, the MS detection of K145 acetylation was done with ΔNLS mutant TDP-43[17,19] lacking the K84 acetylation site we detected with wtTDP-43. It is possible that differential lysine acetylation events occur in cytosolic and nuclear TDP-43 pools. Antibodies directed against [acK145]TDP-43 stained neuropathological inclusions in

ALS but not FTLD-TDP[17]. Yet, MS analysis of TDP-43 PTMs in two ALS cases failed to confirm all of these lysine acetylations but instead found K82 acetylation in one ALS case[45]. The human disease relevance for lysine-acetylated TDP-43 remains to be further explored.

## Methods

**Antibodies.** In this study the following antibodies were used for western blot (WB) and IF staining: rat anti-[acK84]TDP-43 (WB, 1:10 supernatant; IF,1:2; this work), rat anti-[acK136]TDP-43 (WB, 1:10 supernatant; IF,1:2; this work; mouse anti-6xHis (WB, 1:10000; IF, 1:1000; Amersham 27-4710-01), rabbit pan-acetylated lysine (WB, 1:1000; Cell Signalling #9441), rabbit mAb mix pan-acetylated lysine (WB, 1:1000; Ac-K-100, Cell Signalling #9841), mouse anti-Flag HRP-coupled (WB, 1:10000; Sigma #A8592), mouse anti-GAPDH (WB, 1:50000; clone 6C5, Biodesign International #H86504M), rabbit anti-Hsp90 (WB, 1:1000; Cell Signalling #4874), rat anti-[pS409/410]TDP-43 (WB, 1:10; IF, 1:50; clone 1D3), rabbit anti-TDP-43 (WB, 1:8000; IF, 1:1000; ProteinTech 10782-2-AP), mouse anti-TDP-43 (WB, 1:2000; IF, 1:1000; Abnova #H00023435), mouse anti-hnRNPA1 (WB, 1:2000; clone R196), Cell Signalling #5380), mouse anti-tubulin (WB, 1:10000; Sigma #T-5168), mouse anti-ubiquitin (WB, 1:4000; Millipore #MAB1510), mouse anti-YY1 (WB, 1:2000; clone H-10, Santa Cruz Biotechnology #7341), mouse anti-β-actin (WB, 1:50000; clone AC-15, Sigma #A5441); rabbit anti-mouse HRP-coupled (WB, 1:10000, Amersham #NIF825); goat anti-rabbit HRP-coupled (WB, 1:10000, Amersham #NIF824); donkey anti-rat HRP-coupled (WB, 1:10000, Jackson Immunoresearch, 712-035-150) and secondary Alexa Fluor 488- (anti-Mouse: Cat # A32723; anti-Rabbit: Cat # A32731), 555- (anti-Mouse: Cat # A32727; anti-Rabbit: Cat # A32732), 647- (anti-Mouse: Cat # A32728; anti-Rabbit: Cat # A32733) conjugated antibodies for IF produced in goat were from Invitrogen (1:1000).

**cDNA constructs.** wtTDP-43 was cloned into pCMV 5′6His (Clontech) via SalI/NotI. Acetyl-mimic glutamine and acetyl-dead arginine substitutions were introduced by site-directed mutagenesis and cloned into pCMV 5′6His (SalI/NotI). The cloning and mutagenesis primers are listed in Supplementary Table 2. The mutations were introduced via a two-step PCR mutagenesis, where TDP-43 gene was amplified in two fragments: from the start of the gene to the mutant codon, and from the mutant codon to the end. These two fragments were used as a template for a third PCR reaction in this case amplifying the whole length of TDP-43 and introducing restriction sites at both ends.

For the EGFP-tagged constructs, pEGFP c1 wtTDP-43 was used as a template[46]. K136R and K136Q mutations were introduced via site-directed mutagenesis and cloned into pEGFP c1 via BamHI and HindIII.

For the C-terminally tagged amber suppression constructs, TDP-43 was cloned from the pCMV 5′ 6His construct previously mentioned into a pCMV 3′6His vector via SalI and Bsd120I. The amber stop codons from constructs K84TAG and K136TAG TDP-43 were introduced via 2-step point mutagenesis. For the N-terminally tagged amber suppression constructs, TDP-43 was cloned from the amber suppression C-terminally tagged constructs previously mentioned into the vector E400 via NheI and BamHI. The plasmids E400_pAS1_4x7SKPylT_EF1_In_IRES_Bsd (E400) (https://www.addgene.org/140008/; https://benchling.com/s/seq-ZKWKXmPrKBdp8jgpWEt5) and pPB_4xPylT_EF1_IPYE_chAcKRS_Puro (E451) (https://benchling.com/s/seq-6fpczrM3tMNE9fZ0P2nk). Constructs lacking the C-terminal part and with mutated NLS were in addition to K136TAG were cloned from previously described plasmids[6] and cloned into pCMV 3′6His vector via SalI and Bsd120I. Cloning primers are listed in Supplementary Table 2.

Plasmids containing sirtuins were ordered from Addgene, and all of them were produced by Eric Verdin. The plasmids ordered were SIRT1 Flag (Addgene plasmid # 13812; RRID:Addgene_13812), SIRT2 Flag (Addgene plasmid # 13813; RRID:Addgene_13813), SIRT3 Flag (Addgene plasmid # 13814; RRID: Addgene_13814), SIRT4 Flag (Addgene plasmid # 13815; RRID:Addgene_13815), SIRT5 Flag (Addgene plasmid # 13816; RRID:Addgene_13816), SIRT6 Flag (Addgene plasmid # 13817; RRID:Addgene_13817), and SIRT7 Flag (Addgene plasmid # 13818; RRID:Addgene_13818). All sirtuin plasmids were characterized by North et al.[47].

Plasmids containing HDACs 2–5, 7 and 8 were ordered from Addgene. The plasmids ordered were HDAC2 Flag (Addgene plasmid # 36829; http://n2t.net/addgene:36829; RRID:Addgene_36829), HDAC3 Flag (Addgene plasmid # 13819; http://n2t.net/addgene:13819; RRID:Addgene_13819), HDAC4 Flag (Addgene plasmid # 13821; http://n2t.net/addgene:13821; RRID:Addgene_13821), HDAC5 Flag (Addgene plasmid # 13822; http://n2t.net/addgene:13822; RRID:Addgene_13822), and HDAC7 (Addgene plasmid # 13824; http://n2t.net/addgene:13824; RRID:Addgene_13824). HDAC2 plasmid was described by Reyon et al.[48], HDAC3 plasmid was characterized by Emiliani et al.[49] and HDAC4, 5, 7 and 8 were characterized by Fischle et al.[50], HDAC1 and HDAC6 plasmids were generated by Fiesel et al.[29].

**Cell culture and transfections.** HEK293E (ATCC) cells were cultured in Dulbecco's Modified Eagle Medium (DMEM) containing 10% foetal bovine serum (FBS), at 37 °C in 5% CO2. Stable TDP-43 knockdown (shTDP-43 HEK293E) cells were generated and characterized previously[29]. Cells were plated at different concentrations corresponding to the surface of the plate for different experiments. Transfection of plasmids was done 24 h after plating using FuGENE6 (Promega) at a DNA/FuGENE ratio of 1:4.5 according to the manufacturer's instructions.

**Amber suppression.** HEK293E cells were plated and cultured in DMEM with 10% FBS and transfected with FuGENE6 (Promega) 24 h later. Twenty four or 48 h after transfection the media was substituted with DMEM with 10%FBS and 5 mM Nε-Acetyl-L-lysine (A4021, Sigma). Cells were lysed or fixed after being 24 h in the presence of acetylated lysine. When mentioned, Ex527 (CAS 49843-98-3, Santa Cruz Biotechnology) was dissolved in DMSO and added to the cultured cells 24 h before lysis to a final concentration of 1 or 10 μM.

**Nuclear-cytoplasmic fractionation protocols.** For the nuclear-cytoplasmic fractionation shown in Supplementary Fig. 1b, shTDP-43 HEK293E cell pellets from confluent 10 cm plates were washed twice in ice-cold PBS and then resuspended in 500 μl of hypotonic buffer A (10 mM HEPES, 1.5 mM MgCl2, 10 mM KCl, 0.1 mM DTT and protein inhibitor cocktail (Roche) and incubated on ice for 5 min. Cell membranes were ruptured with 20 strokes in an ice-cold Dounce homogenisator and resulting samples were centrifuged at 500 × g for 5 min at 4 °C. Supernatants (cytosolic fraction) were mixed with 5× RIPA buffer, incubated for 10 min on ice and cleared by spinning the samples at 5000 × g for 15 min at 4 °C. Pellets containing nuclei and residual cytoplasmic proteins were washed twice with hypotonic buffer A. After resuspension in 500 μl S1 buffer (250 mM sucrose and MgCl2), samples were layered carefully over 500 μl S3 buffer (880 mM sucrose and MgCl2) and centrifuged at 3000 × g, for 10 min at 4 °C. The supernatant was discarded, and the remaining pellet was lysed with urea buffer. DNA was sheared by passing lysates 30 times through a 23-gauge needle. The cleared supernatants in loading buffer were analyzed via Western blotting.

For the nuclear-cytoplasmic fractionations of HEK293E cells shown in Supplementary Fig. 1d, the Subcellular Protein Fractionation Kit for Cultured Cells (Cat. no: 78840, Thermo Fisher) was used according to manufacturer instructions. Fractions were processed for Western blotting as described above. Ten micrograms of lysate fraction F1 (soluble cytoplasmic) was loaded, and 3 μg each was used for fractions F2 (insoluble/membrane-bound), F3 (soluble nuclear) and F4 (chromatin-bound).

**Cell lysis, solubility fractionation, and western blotting.** Cells were collected with a cell scraper and lysed in urea lysis buffer (10 mM Tris pH 8, 100 mM NaH2PO4, 8 M urea). For the sequential extraction cells were first lysed with RIPA buffer (50 mM Tris/HCl pH 8, 150 mM NaCl, 1% NP-40, 0.5% deoxycholate, 0.1% SDS, 10 mM sodium pyrophosphate) with proteinase inhibitor cocktail (Roche). Samples were centrifuged at 5000 × g for 15 min, leaving the RIPA-soluble proteins in the supernatant. The remaining pellet was then lysed in urea buffer, and insoluble material was removed with a centrifugation step at 5000 × g for 15 min. DNA was sheared with a 23-gauge needle. Protein concentration in lysates in urea buffer was quantified with a Bradford Protein assay kit (Biorad) and lysates in RIPA buffer with BCA protein assay kit (Pierce). 3x Laemmli loading buffer with 100 mM DTT was added to the samples and afterwards they were boiled at 95° for 5 min. The denatured samples were then subjected to western blot analysis. Denatured samples were loaded and run in polyacrylamide gels. The gels were then blotted using the wet-transfer Trans-Blot® Cell system from Bio-Rad. Proteins were transferred to Hybond-P polyvinylidene difluoride membranes (Millipore). Membranes were blocked with 5% non-fat milk/TBS-T for an hour at room temperature (RT). Primary antibody incubation took place overnight at 4 °C. The membranes were washed and incubated with HRP-coupled secondary antibody for 1–2 h at room temperature. Membranes were washed and proteins were detected with Immobilon Western chemiluminescent HRP substrate (Millipore) using the ChemiDoc XRS+ Imaging System.

**Generation of monoclonal antibodies against lysine acetylated TDP-43.** Two female Lou/c rats aged 12 weeks were immunized (approval by Regierung von Oberbayern AZ 55.2-1-54-2531.6-4-99) with 2 ovalbumin-coupled synthetic peptides at the same time: one with amino acids corresponding to residues 79-89 of TDP-43, with K84 acetylated (C-KDNKR(Ac)KMDETD); and the second one corresponding to residues 131-141, with K136 acetylated (C-LMVQV(Ac)KKDLKT). Animals were sacrificed and splenocytes were fused with mouse myeloma cells. Hybridoma supernatants were tested for binding to acetylated and non-acetylated peptides by ELISA. Those supernatants that were positive for the acetylated peptides and negative for the non-acetylated peptides were further validated on lysates of cells expressing either wt or amber-suppressed TDP-43 by Western blot analysis (dilution 1:10) and IF staining (dilution 1:2). Hybridoma clones TDACA 9G9 (IgG2c/κ) against [acK84]TDP-43, TDACB 23B3 (IgG1/κ) (used only in supplementary Fig. 4) and TDACB 14D4 (IgG2b/κ) against [acK136] TDP-43 were stably established by limiting dilution cloning.

**Immunofluorescence staining.** Cells were plated in 6-well plates and grown for 24 h. They were then transfected and 6 h after transfection they were transferred

(diluted 1:10) to coverslips in a 24-well plate coated with poly-D-lysine (PDL) and collagen. Cells then grew for 24–72 h before fixation with 4% PFA/PBS for 20 min at RT depending on the experiment. Permeabilization of the cell membrane was done with 1% Triton-X-100/PBS for 5 min at RT. Blocking was done with 10% Normal Goat Serum in PBS, for 1 h at RT. The primary antibody incubations were done at the already mentioned concentrations, in 1%BSA/PBS for 2 h at RT. The secondary antibody incubation was done in 1% BSA/PBS for 1 h. Mounting was done using 40 μl of Dako mounting media per slide, following a 4° incubation overnight. Immunofluorescence images were acquired using a Zeiss fluorescence microscope with Apotome attachment (Axio imager z1 stand).

**Fluorescence recovery after photobleaching.** HEK293E cells were plated on glass bottom chambered sides (Lab Tek, 154526) coated with PDL and collagen and transfected with C-terminally EGFP-tagged wt or K136Q TDP-43 after 24 h. Imaging and photobleaching was done with a Zeiss LSM510 META confocal microscope, using a x63/1.4 oil objective. ROIs were bleached for 1 s using a 405 nm and a 488 nm laser at maximum power output. Cells were continuously imaged for 3 min after bleaching. Reference ROIs were measured from unbleached cells to correct for photobleaching due to fluorescence imaging.

**Live cell imaging.** HEK293E cells were plated on glass-bottom chambered slides coated with PDL. They were transfected with C-terminally EGFP-tagged wt or K136Q TDP-43 after 24 h. While imaging cells were kept in a temperature-controlled chamber at 37° with 5% $CO_2$. Imaging was done with an Axio observer Z1.

**RNA-protein pulldown.** High-performance liquid chromatography purified RNA oligonucleotides $5'$-$(UG)_{12}$-$3'$ and $5'$-$(UC)_{12}$-$3'$ were ordered from Sigma. The oligonucleotides were biotinylated using the Pierce RNA $3'$ End Desthiobiotinylation kit (Cat no 20163) with small variations. For each 1nmol of biotinylated cytidine bisphosphate 100 pmol of RNA was added to the biotinylation reaction. Biotinylated RNA oligos were purified using chloroform:isoamyl alcohol. For the RNA-protein pulldown the Pierce Magnetic RNA-Protein Pulldown kit (Cat no 20164) was used with small variations. Cells were lysed in NP-40 buffer (50 mM $NaH_2PO_4$ pH 8.0; 300 mM NaCl; 1% NP-40) and 400 μg of protein was used for each RNA-protein binding reaction. After 1 h incubation proteins were eluted in 25 μl Laemmli buffer.

**Native protein pulldown and filter-binding assay.** HEK293E sh[TDP-43] cells were plated in 10 cm plates. After 24 h, cells were transfected with the respective plasmids. After 48 h, cells were lysed in Native NiNTA lysis buffer (50 mM $NaH_2PO_4$, 300 mM NaCl, pH 8, 10 mM imidazole) and protein concentration was measured via Bradford assay (Bio-Rad). 500-800 μg of protein were incubated with NiNTA beads at 4 °C overnight. Beads were washed with Native NiNTA lysis buffer with 20 mM imidazole. After washing, proteins were eluted in Binding buffer (10 mM HEPES, 20 mM KCl, 1 mM $MgCl_2$, 1 mM DTT, 5% glycerol) containing 300 mM imidazole, which was then quantified measuring the absorbance at 280 nm using a Nanodrop 2000 (Thermo Fisher Scientific). The purified protein was incubated at various concentrations with 2 nM of biotinylated poly-$(UG)_{12}$ RNA for 30 min at room temperature. The resulting complex was run through a Bio-Dot® SF Microfiltration apparatus (Bio Rad, Cat no: 170-6542) loaded with a nitrocellulose and a positively charged nylon membrane. Membranes were then cross-linked at 120 J/m² and then labelled RNA was detected using the Chemiluminescent Nucleic Acid Detection module kit (Thermofisher, cat no: 89880) according to the manufacturer's instructions. In addition, for quantification purposes pulldown fractions were analyzed via SDS-PAGE electrophoresis and further Western blot. Band intensity corresponding to wtTDP-43 and K136R TDP-43 was quantified using ImageJ 1.52a and the resulting data was used to normalized the Dot Blot densitometry.

**Nickel pulldown of 6xHis-tagged TDP-43.** Cells were plated in 10 cm plates for 24 h before transfection. Seventy-two hours after transfection cells were lysed in urea lysis buffer containing 10 mM imidazole. DNA was sheared with a 23-gauge needle. After quantifying the amount of protein in the lysates with a Bradford Protein assay kit (Biorad), 500–800 μg of protein were incubated with NiNTA agarose beads (Qiagen) at 4 °C overnight. Beads were washed with Urea wash buffer (10 mM Tris pH 6.3, 100 mM NaH2PO4, 8 M urea) containing 20 mM imidazole. Afterwards proteins were eluted in 3x Laemmli buffer at 95° for 10 min. Eluted proteins were analyzed in Western blot.

**RNA extraction and RT-PCR.** HEK293E sh[TDP-43] cells were plated in 6-well plates and after 24 h were double-transfected with the corresponding TDP-43 construct and a CFTR minigene construct[11] at a ratio of 2:1. After 72 h RNA was isolated using the RNeasy Mini Kit (Qiagen) following the indications from the provider. From the isolated RNA cDNA was obtained using the Transcriptor High Fidelity cDNA Synthesis Kit (Sigma) following the protocol of the provider. The cDNA of interest was then amplified via a standard PCR and the product was run in a 2% agarose gel. DNA was visualized using Midori Green from Biozym and the ChemiDoc XRS + Imaging System.

**Statistical analyses and molecular representations.** Cellprofiler 4.1.3 was used for quantification of IF pictures. Statistical significance was calculated using Microsoft Excel 2016 and Prism 9.3.0 (Graphpad). Western blot images were captured with ImageLab 5.0. Densitometry of electrophoresis gels was performed using Imagej 1.52a (NIH). P values below 0.01 were considered significant. Unpaired t-tests were performed to compare two variables and assess significance. Chi-squared test was used to assess significance in the IF experiments. n values detailed in each figure legend. Graphical representations of TDP-43 mutants and acetylated forms was made with Pymol (version 2.4.0). Post translational modifications were modulated using the PyTMs plugin by Warnecke et al.[51]

**Reporting summary.** Further information on research design is available in the Nature Research Reporting Summary linked to this article.

## Data availability

All data generated for this study are provided and can be found in the Supplementary Data and Source Data files found together with this paper. The mass spectrometry proteomics data have been deposited to the ProteomeXchange Consortium via the PRIDE[52] partner repository with the dataset identifier PXD030170. Source data are provided with this paper.

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

## Acknowledgements

We would like to thank Dr. Ivana Nikić-Spiegel and Dirk Schwarzer for helpful discussions, Dr. Petra Frick for the validation of the acetyl-TDP-43 antibodies, Dr. Sven Geisler for excellent technical advice, Stefan Hauser for help with Live Cell Imaging, and Anna Lechado Terradas for critical comments on the manuscript. This work was supported by the NOMIS Foundation (P.J.K. and M.N.); German Research Foundation (DFG) grant KA1675/3-2 (P.J.K.); the German Center for Neurodegenerative Diseases (DZNE) within the Helmholtz Association (P.J.K., M.N., C.J.G., R.F.); the Hertie Foundation (P.J.K.); and the AriSLA PathensTDP project (E.B.).

## Author contributions

J.G.M. designed and performed most experiments and analyzed the results. F.H. assisted the study and provided TDP-43 reagents. F.v.Z. and C.J.G. performed MS and analyzed the data. R.F. generated the antibodies against acetylated TDP-43. S.J.E. provided the amber suppression methodology and reagents. A.A.S. established the FRAP assay. E.B. provided the CFTR reporter minigene and methodological support for splicing and RNA binding assays. M.N. contributed to the study design and provided neuropathological tools. M.N. and P.J.K. acquired funding. P.J.K. conceptualized and supervised the study. J.G.M. and P.J.K. wrote the draft and all co-authors edited the paper.

## Funding

## Competing interests

The authors declare no competing interests.
