## [Peer Review File · Nature Communications]

Reviewers' comments:

Reviewer #1 (Remarks to the Author):

The manuscript describes a novel regulatory pathway of TDP-43 posttranslational modification regulating the nuclear import and RNA binding activity of TDP-43. These processes are important for TDP-43 normal function and are relevant to neurodegenerative disease mechanisms, mainly ALS and frontotemporal dementia, associated with the protein. The authors nicely show that they can express acetylated TDP-43 in cells, showing similar behavior as the K136 and K84 mimics. Moreover, they developed acK-specific antibodies that detect acK84 and acK136 in their cell model. These findings provide insight into new posttranslational modifications controlling the activity of TDP-43 and point to new potential therapeutic targets and diagnostic markers for disease. However, the experimental design to characterize the role of K84 and K136 modifications on TDP-43 function are not fully convincing. In addition, the data to support the regulation of K136 acetylation by sirtuins is not strong. Therefore, although the manuscript describes interesting new findings on the regulation of TDP-43, it does not provide sufficient data to support the conclusions.

Major points

1. One major issue with assessing the effect of mutations on TDP-43 function by transient transfection of WT and mutant constructs is that the levels of expression are widely heterogeneous and often leads to overexpression, which in turn greatly alters cellular dynamics and solubility. This also applies in the case of the Lys mutants characterized for this work, as shown in Fig 1. To address this complication, TDP-43 studies should provide evidence for close to endogenous and homogeneous levels of overexpression.
2. Related to point 1 above, the levels of expression of different mutants are widely different as shown in Fig 1D for Δ NLS and Figs 2A and 2E for FLL. There seems to be no expression of FLL in the expt shown in Fig 2E. In addition, the MW for FLL in Figs 2A and 2E appears greater than the other TDP-43 constructs. This is not addressed in the text.
3. Figs 1B and 1C show that K84 is mostly cytoplasmic. However, nuclear/cytoplasmic fractionation in Fig 1D show mostly nuclear K84 localization. Comparison with the Δ NLS as control would have been helpful to clarify this discrepancy, but the expression of this variant is minimal (Fig 1D).
4. The authors propose that acetylation at K136 causes changes in cellular dynamics and LLPS properties as a result of decreased RNA binding. In addition to the IP results using RNA to pull down WT and K136Q, the experiments should include measurements of RNA binding affinity to compare WT and K136 mutants. Fig 2D should also include IP of the FLL mutant as control.
5. The plot of Fig 2F does not make sense as it seems that the y axis is wrongly labeled. In addition, this quantification refers to expts shown in Fig 2E and the plot should include the same variants, why were only some selected and did not include all? To explain why K84Q does not affect splicing if the expression of the mutant is mostly cytoplasmic as shown in Fig 1B, Δ NLS should be used as control.
6. Fig 6A showing acK136 detection needs to be replaced with an experiment/image of better quality because the first lane control band is covered by a large dark spot. The additional band of higher MW found in all lanes is not discussed by the authors. The sirtuin expression levels detected by the Flag antibody vary widely where Sirt1, 3 and 4 are almost undetectable. These results do not provide strong evidence that a specific enzyme, and not others, acts on acK136. In addition, Fig 6B is difficult to interpret due to the lack of obvious differences or quantification. In all, the data do not support the conclusions regarding a possible role of sirtuins in regulating acK136 and the abstract stating that "SIRT1 can potentially deacetylate [acK136]TDP-43" needs to be revised accordingly.
7. Supp Fig 2A does not make sense. The labels for the different lanes do not correspond to the bands and all the bands are cut off.

Minor points

1. Several references to previously published findings on TDP-43 are missing.
Pg 2 lane 42 Ref to Ayala YM et al. 2008 should be added for nuclear cytoplasmic shuttling.
Pg 2 lane 42 Ref to Tollervey JR 2011 should be added for UG-rich RNA binding.

Pg 5 Lane 106 Ref to F147L/F149L cellular localization Ayala YM et al 2008 is missing.

Reference #9 on pg 19, lane 490 is wrong.

2. Reference to the NES in Fig 1A diagram of TDP-43 domain organization should be removed as recent evidence from different groups has shown that this NES is not a true export signal. In the same figure, to help readers, the residue positions of the acetylation sites should be added on top of the arrows.

3. Fig 2G is confusing and hard to interpret. K136 should be labeled and atoms belonging to either protein or RNA may be colored differently to help visualize contacts.

Reviewer #2 (Remarks to the Author):

Morato and colleagues report here a role for site-specific acetylation in the regulation of the well-studied DNA/RNA-binding protein TDP-43.

Previous published work on TDP-43 highlighted the ability of this nuclear protein to shuttle between the nucleus and the cytoplasm and also to phase separate and to form insoluble aggregates.

Here the authors demonstrate that the acetylation of two specific lysines, K48 and K136, could control the already published properties and activities of the protein and identified SIRT1 as a modulator of the functions of TDP-43 depending on K136 acetylation, by specifically deacetylating this site.

Overall, the new information provided here, although possibly of interest to the specialists of TDP-43, does not represent a novelty in this field that could be of interest to a large audience.

Additionally, no relevant physio-pathological system was used to demonstrate the occurrence of the regulation of TDP-43 by these site-specific acetylations; all the conclusions remain based on totally artificial systems.

Finally, there are many ambiguous experimental results which do not support the clear-cut authors' conclusions.

Specific points

1 - Figure 2D: the quality of the immunoblot does not allow the authors to claim that the acetyl-mimic TDP-43 K136Q mutant shows a reduction in RNA-binding.

The TDP-43 K136Q band appears as fuzzy and migrates as a doublet.

2 - Why is TDP-43 K136R mutant not used in this assay?

3 - If, as discussed in lines 155-161, the structural role of K136 is true then it would be difficult to present the K136Q and K136R substitutions respectively as the acetyl-mimic and the acetyl-dead mutants. Indeed, the effects observed with these mutants could merely reflect defects in the structure of the protein rather than a role for acetylation / non-acetylation of this residue.

4 - The data corresponding to the FRAP experiments (Figure 4) should be presented in a more professional way by indicating $t_{1/2}$, mobile fractions, ...

5 - Figure 5, why does the anti-K136ac antibody detect an additional band at around 37kDa? Why do both the TDP-43 band and this unknown band decrease in the K136-TAG lane?

The same question stands for K84ac.

Given the fact that the anti-K136ac detects a strong band in all samples by immunoblot (Fig. 5A), why would this antibody only detect K136-TAG expressing cells by immunofluorescence (Fig. 5C)?

6 - It is not clear why, among the three classes of histone/protein deacetylases, the authors choose to focus on the sirtuins. The explanation that sirtuins are involved in neurodegenerative diseases does not justify the lack of interest in the other deacetylases.

7 - Figure 6A shows that the expression levels of the tested sirtuins are very variable, making it very difficult to conclude on the comparative activities of these enzymes. Additionally, the K136ac signal shown correlates better with the total amount of TDP-43 than with the expression of the tested sirtuins.

8 - The selection of only two cells shown in Figure 6B is not enough to conclude on the role of sirtuins in the control of protein aggregates formation. Additionally, these data question the role of Sirt1 in the deacetylation of TDP-43 K136, since no reduction of the K136ac signal is observed in SirT1 co-expressing cells.

Reviewer #3 (Remarks to the Author):

The work of Morato et al. focuses on the effect of acetylation on the cellular localization and phase-separation propensity of TDP-43, a critical protein involved in ALS and FTD. The authors report that K84 acetylation reduces the nuclear import, whereas K136 acetylation impaired RNA binding and splicing activity. The failure of RNA interaction is proposed to lead to a phase-separation of the protein – directed by the C-terminal - and formation of insoluble aggregates that have characteristic similar to the pathological ones in terms of the phosphorylation and ubiquitination of the protein. Morato et al. makes use of an elegant amber suppression to introduce “authentic” acetyl-lysine in the protein sequence and further confirm the observations obtained with site-directed mutagenesis. Finally, they develop new specific antibodies and identify deacetylation of K136 by SIRT1. Overall, these results are fascinating and paint a picture where the strong regulation of post-translational modifications alter the localization and interaction of TDP-43 with other components. In this respect, the manuscript of Morato et al follows similar moves than the ones proposed by Cohen et al. (ref. 17) and provides corroboration that acetylations act as a switch for function and aggregation. The investigation of PTMs is fundamental for understanding the interplay between TDP-43, phase-separation, and neurotoxicity. However, some points in the manuscript raises concerns that affect the enthusiasm for the reported findings and possibly limit the interest of the manuscript for a broad audience.

1. The authors do not provide a strong motivation for the investigation of residue acetylation and its possible role in modulating cellular localization. Findings appear somehow circumstantial “While investigating the sites of TDP-43 lysine ubiquitinations, we detected in the mass spectrometry another lysine modification, namely acetylation” (80-81). A stronger driving hypothesis, at least on the role of specific mutations and how they can affect TDP-43 localization, in the light of previous models, would help clarifying the motivation of the work. At the same time, a schematic model of interactions summarizing the discovered modulation of PTMs and their relation to demixing may help influencing the field.

2. At line 66. the authors state that “we discovered that acetylation of K84 within the nuclear localization signal reduced nuclear import whereas acetylation of K136 in the RNA recognition domain impaired TDP-43 binding and splicing capabilities”. However, it was previously reported in literature that alterations of K84 can affect nuclear localization (e.g. Jiang et al, Scientific Report, 2017 use alterations in lysine and arginine residues within the NLS to suppress nuclear localization) and different localization and phase-separation of K136 mutants have been shown in ref. 16 (but not really discussed in the manuscript). A better contextualization of previous experiments and a critical indication on how specific mutations may affect the protein can help clarifying the original findings in the manuscript. One important advancement remains the testing of interactions of the protein with nucleic acids.

3. The authors often use the term “aggregates” and the phase-separation terminology across the text is somehow confusing, since a phase separated granule is not an aggregate (perhaps it is an aggregation of multiple components, but the specific interaction at play usually confer a certain degree of fluidity, very specific concentration boundaries (binodal), and reversibility). The aggregate term

usually invokes more static and possibly unspecific and/or irreversible formations. The manuscript may benefit of a better usage of nomenclature in this respect.

4. The nuclear "aggregates" of TDP-43 K136R mutant have a different distribution and size that does reported in ref. 16, where large "droplets" are observed. Can the author comment on this difference? Technically speaking, this is an important element in judging whether the protein is undergoing phase-separation or forming oligomers.

5. On a similar line, the authors report in Fig. 4 FRAP experiments from objects that are larger than 1 um and objects that are smaller than 0.5 um. A more detailed description of the experiments and choice of ROI would help the reader in contextualizing the experiments. Can the different velocity and amplitude of recover be related to the size of the objects? Or is there any clear evidence supporting that small "puncta" have physical properties different from "larger" ones (e.g. ones fusing, the other ones not)? The authors mention also "droplets fusing" at line 214-215. Can the authors quantify whether the average size of droplets is increasing over time (another quality of phase-separating systems) and whether there is a size threshold or time threshold that limit the growth? Have the authors tested FRAP across different delay times after droplets are formed to see whether recovery is time dependent (which would follow the hypothesis described in lines 216-217)?

6. Have the authors tested the effect of temperature to verify if there is any transition in the solubilization of these puncta as a proxy for confirming the phase-separation nature and reversibility of their assembly vs. aggregation of the protein?

Reviewer #4 (Remarks to the Author):

Morato et al., describes that acetylation at sites of Lysine 84 and Lysine 136 of TDP-43 regulates its cytoplasmic mislocalization and the aggregation propensity. Deacetylase SIRT1 is responsible for deacetylation of TDP-43 at Lysine 136 and reduces the aggregation propensity of TDP-43. The strengths of manuscript include successfully incorporating authentic acetyl-lysine at the apparently regulatory sites in TDP-43 with amber suppression methodology. Overall, the data provided in this manuscript are convincing and with proper controls. However, there are a couple of concerns, which are listed below.

1) In Figure 1D, the bands for TDP-43 should be quantified by densitometry.

2) In Figure 2D, both mutants, K136R and K136Q, increase ubiquitination and phosphorylation of TDP-43. The wild-type TDP-43 and authentic acetylation of K136 of TDP-43 should be compared for their ubiquitination and phosphorylation status to verify the crosstalk among acetylation, ubiquitination, and phosphorylation of TDP-43. The resulting data should be discussed in the discussion section.

Reviewer #1

The manuscript describes a novel regulatory pathway of TDP-43 posttranslational modification regulating the nuclear import and RNA binding activity of TDP-43. These processes are important for TDP-43 normal function and are relevant to neurodegenerative disease mechanisms, mainly ALS and frontotemporal dementia, associated with the protein. The authors nicely show that they can express acetylated TDP-43 in cells, showing similar behavior as the K136 and K84 mimics. Moreover, they developed acK-specific antibodies that detect acK84 and acK136 in their cell model. These findings provide insight into new posttranslational modifications controlling the activity of TDP-43 and point to new potential therapeutic targets and diagnostic markers for disease.

We thank the reviewer for the positive reception of our findings.

Major comments:

1. One major issue with assessing the effect of mutations on TDP-43 function by transient transfection of WT and mutant constructs is that the levels of expression are widely heterogeneous and often leads to overexpression, which in turn greatly alters cellular dynamics and solubility. This also applies in the case of the Lys mutants characterized for this work, as shown in Fig 1. To address this complication, TDP-43 studies should provide evidence for close to endogenous and homogeneous levels of overexpression.

We agree that uneven TDP-43 overexpression can be a confounding factor. To avoid these artifacts, we always compare transfected TDP-43 mutants in parallel with wtTDP-43 in a sh^{TDP-43} depleted context. In this controlled manner, we determine the impact of the mutations or PTMs with a similar level of expression. In addition, due to the inefficiency of the system, amber suppressed TDP-43 has lower expression levels than transfected wtTDP-43 (Fig. 4a), closer to the levels of endogenous TDP-43. Taken these two points in account, we are confident that the observed effects can be ascribed to the TDP-43 PTMs.

2. Related to point 1 above, the levels of expression of different mutants are widely different as shown in Fig 1D for Δ NLS and Figs 2A and 2E for FFLL. There seems to be no expression of FFLL in the expt shown in Fig 2E. In addition, the MW for FFLL in Figs 2A and 2E appears greater than the other TDP-43 constructs. This is not addressed in the text.

We have now optimized the expression levels of both Δ NLS and FFLL (cf new Fig. 1d,e). For the important splicing experiment, we subcloned the FFLL construct into the 6xHis tag vector, achieving similar expression levels and electrophoretic motilities (new Fig. 3e). We have clarified in the legend for Fig. 2a that the cause of the size difference is due to a 3xFlag tag in the FFLL construct versus 6xHis tag in the other constructs.

3. Figs 1B and 1C show that K84 is mostly cytoplasmic. However, nuclear/cytoplasmic fractionation in Fig 1D show mostly nuclear K84 localization. Comparison with the Δ NLS as control would have been helpful to clarify this discrepancy, but the expression of this variant is minimal (Fig 1D).

We realize this impression came from pictures of cells with extreme cytosolic labeling of [K48Q]TDP-43 in the previous Fig. 1b, which is now replaced by a more representative image. Though nearly all transfected cells show cytosolic mislocalization of [K84Q]TDP-43 (clarified in the quantification Fig. 1c), some protein remains in the nucleus, consistent with the biochemical fractionation experiments and splicing assays.

Please note that we see a similar behavior for the established Δ NLS construct (revised Fig. 1d with better matched expression levels). While there is significantly enhanced cytoplasmic to nuclear distribution (new quantification Fig. 1e), detectable amounts of nuclear import impaired TDP-43 can be found in the nucleus of our experimental system (see also Hans et al. 2018). As explicated in the revised text, our modifications involving the NLS K84 reduce nuclear import but do not completely abolish it.

4. The authors propose that acetylation at K136 causes changes in cellular dynamics and LLPS properties as a result of decreased RNA binding. In addition to the IP results using RNA to pull down WT and K136Q, the experiments should include measurements of RNA binding affinity to compare WT and K136 mutants. Fig 2D should also include IP of the FFL mutant as control.

For better quantification of the RNA binding properties, we have developed a filter-binding assay, titrating TDP-43 to biotinylated UG₁₂ (new Fig. 3b-d). This new quantitative assay confirms the previous qualitative pulldown result (now Fig. 3a).

5. The plot of Fig 2F does not make sense as it seems that the y axis is wrongly labeled. In addition, this quantification refers to expts shown in Fig 2E and the plot should include the same variants, why were only some selected and did not include all? To explain why K84Q does not affect splicing if the expression of the mutant is mostly cytoplasmic as shown in Fig 1B, Δ NLS should be used as control.

As correctly pointed out by the reviewer, the y-label was wrong and it is now corrected (revised Fig. 3f). We have now included the quantification of splicing activities of all the variants. As requested, we have added the Δ NLS construct as a control and indeed found that the splicing capabilities of K84Q TDP-43 are similar to those of Δ NLS TDP-43. Nevertheless, both rescue CFTR splicing to a lesser extent than wtTDP-43. Thus, the altered cytoplasmic to nuclear ratio (Fig. 1e) of the nuclear import impaired TDP-43 variants reduces their splicing potential (Fig. 3e,f).

6. Fig 6A showing acK136 detection needs to be replaced with an experiment/image of better quality because the first lane control band is covered by a large dark spot. The additional band of higher MW found in all lanes is not discussed by the authors. The sirtuin expression levels detected by the Flag antibody vary widely where Sirt1, 3 and 4 are almost undetectable. These results do not provide strong evidence that a specific enzyme, and not others, acts on acK136. In addition, Fig 6B is difficult to interpret due to the lack of obvious differences or quantification. In all, the data do not support the conclusions regarding a possible role of sirtuins in regulating acK136 and the abstract stating that “SIRT1 can potently deacetylate [acK136]TDP-43” needs to be revised accordingly.

We have developed this point considerably with new experimentation. The original survey Fig. 6a was moved into the supplement (Fig. S4c). A new Fig. 7a clearly shows the [acK136]TDP-43 deacetylation by SIRT1 transfection. Specificity is confirmed by the dose-dependent inhibition with ex527. By comparison, expression of the matched SIRT2 had no effect. Quantification of the immunofluorescence data is now added (new Fig. 7c). Although we cannot rule out some contribution of other lysine deacetylases (see also new HDAC Fig. S4a,b), we find our data supports the conclusion that “SIRT1 can potently deacetylate [acK136]TDP-43”.

7. Supp Fig 2A does not make sense. The labels for the different lanes do not correspond to the bands and all the bands are cut off.

Sorry, we do not find anything wrong with this figure, which shows the expression levels of the amber-suppressed constructs (now Fig. S3a). Is there a misunderstanding or file conversion error?

In response to the minor comments, we have added the suggested references, removed NES from the TDP-43 scheme Fig. 1a and added the residue numbers, as requested. Fig. 2g was moved into the supplement (now supplementary figure 2) and expanded by more modeling of the residue K136 contacting RNA.

Reviewer #2

Morato and colleagues report here a role for site-specific acetylation in the regulation of the well-studied DNA/RNA-binding protein TDP-43. Previous published work on TDP-43 highlighted the ability of this nuclear protein to shuttle between the nucleus and the cytoplasm and also to phase separate and to form insoluble aggregates. Here the authors demonstrate that the acetylation of two specific lysines, K84 and K136, could control the already published properties and activities of the protein and identified SIRT1 as a modulator of the functions of TDP-43 depending on K136 acetylation, by

specifically deacetylating this site.

Overall, the new information provided here, although possibly of interest to the specialists of TDP-43, does not represent a novelty in this field that could be of interest to a large audience.

Additionally, no relevant physio-pathological system was used to demonstrate the occurrence of the regulation of TDP-43 by these site-specific acetylations; all the conclusions remain based on totally artificial systems.

Finally, there are many ambiguous experimental results which do not support the clear-cut authors' conclusions.

Using the well-established HEK293 cell culture model and a battery of standard assays, we provide validations for two newly discovered TDP-43 PTMs, as summarized in the reviewer's first paragraph. The amber suppression system is certainly artificial, but it allows for the first time site-specific introduction of authentic acetyl-lysine into TDP-43, circumventing problems of amino acid substitutions by conventional mutagenesis. Together with several clarifications, control experiments and quantifications, we believe the revised manuscript sufficiently supports the conclusions that K84 acetylation shifts nucleo-cytoplasmic shuttling towards cytosolic mislocalization, and that K136 acetylation impairs TDP-43 RNA binding, eventually leading to phase separation and aggregation, a process that can be reverted by SIRT1.

Specific points

1 - Figure 2D: the quality of the immunoblot does not allow the authors to claim that the acetyl-mimic TDP-43 K136Q mutant shows a reduction in RNA-binding. The TDP-43 K136Q band appears as fuzzy and migrates as a doublet.

To substantiate the results of the RNA-protein pulldown we have now added a quantitative filter-binding assay, titrating TDP-43 to biotinylated UG₁₂ (new Fig. 3b-d). The results confirm the conclusions from the qualitative pulldown assay (now Fig. 3a).

2 - Why is TDP-43 K136R mutant not used in this assay?

Based on structural considerations (new Fig. S2b), we fear that the K136R mutation causes protein backbone defects. Thus, [K136R]TDP-43 may not be a clean acetylation-dead TDP-43 variant; wtTDP-43 with very low K136 acetylation levels was used in the RNA binding assays in comparison to the "acetyl-mimic" [K136Q]TDP-43. The limitations of the amino acid substitutions are discussed in the manuscript and were overcome in subsequent cellular validation experiments using the amber suppression system.

3 - If, as discussed in lines 155-161, the structural role of K136 is true then it would be difficult to present the K136Q and K136R substitutions respectively as the acetyl-mimic and the acetyl-dead mutants. Indeed, the effects observed with these mutants could

merely reflect defects in the structure of the protein rather than a role for acetylation / non-acetylation of this residue.

We agree with the reviewer completely. The point-mutagenesis approach is a widely used strategy to study PTMs and it was the first approach that we used to characterize the acetylation sites identified by mass spectrometry. We later realized that the method, however, was not adequate for this specific study and turned to amber suppression as a method of choice.

4 - The data corresponding to the FRAP experiments (Figure 4) should be presented in a more professional way by indicating $t_{1/2}$, mobile fractions, ...

The analysis of the FRAP data has been extended with measures of mobile fractions (new Fig. 5c). Due to the extremely low mobility of the big aggregates, measures of $t_{1/2}$ were highly variable. For this reason, we left $t_{1/2}$ measures out of the analysis.

5 - Figure 5, why does the anti-K136ac antibody detect an additional band at around 37kDa? Why do both the TDP-43 band and this unknown band decrease in the K136-TAG lane?

The same question stands for K84ac.

Given the fact that the anti-K136ac detects a strong band in all samples by immunoblot (Fig. 5A), why would this antibody only detect K136-TAG expressing cells by immunofluorescence (Fig. 5C)?

We have established a new monoclonal antibody against acK136 that is more specific and more sensitive than the previous anti-[acK136]TDP-43. The new antibody does not recognize any additional bands in Western blot and it is therefore more reliable. The criticized experiments were repeated with the improved antibody (revised Fig. 6) and fully confirmed our previous results.

6 - It is not clear why, among the three classes of histone/protein deacetylases, the authors choose to focus on the sirtuins. The explanation that sirtuins are involved in neurodegenerative diseases does not justify the lack of interest in the other deacetylases.

Yes, we were also interested in the interaction of other deacetylases with TDP-43. We did look at the effect of HDACs overexpression in K136 acetylation levels. Unfortunately, we did not find a clear effect for any of them, as it is now stated in the corresponding results section. Supplementary figure 4 now shows representative blots with both HDAC and sirtuin coexpression. It is worth noting that HDAC1 and HDAC6 both caused a reduction in TDP-43 levels, preventing us to analyze their impact on K136 acetylation. We hypothesize that these HDACs interfere with the amber suppression system at the tRNA level, however this explanation would require more research, which is currently out of the scope of the paper.

7 - *Figure 6A shows that the expression levels of the tested sirtuins are very variable, making it very difficult to conclude on the comparative activities of these enzymes. Additionally, the K136ac signal shown correlates better with the total amount of TDP-43 than with the expression of the tested sirtuins.*

Right; we moved the original survey Fig. 6a into the supplement (now Fig. S4c). A new Fig. 7a clearly shows the [acK136]TDP-43 deacetylation by SIRT1 transfection, and a selective inhibitor (ex527) prevented SIRT1-mediated deacetylation of [acK136]TDP-43. By comparison, expression of the matched SIRT2 had no effect.

8 - *The selection of only two cells shown in Figure 6B is not enough to conclude on the role of sirtuins in the control of protein aggregates formation. Additionally, these data question the role of Sirt1 in the deacetylation of TDP-43 K136, since no reduction of the K136ac signal is observed in SirT1 co-expressing cells.*

We have now added a quantification for the cells with aggregates in the presence or absence of SIRT1 (new Fig. 7c). Indeed, the acetylation does not disappear completely after SIRT1 overexpression, both in Western blot and immunofluorescence. We think that the remaining acetylation also observed in Western blots (Fig. 7a) is the acetylation that can be detected by immunofluorescence. Taken together, these two experiments suggest a critical amount of acetylated TDP-43 is needed for phase-separation and subsequent aggregate formation.

Reviewer #3

The work of Morato et al. focuses on the effect of acetylation on the cellular localization and phase-separation propensity of TDP-43, a critical protein involved in ALS and FTD. The authors report that K84 acetylation reduces the nuclear import, whereas K136 acetylation impaired RNA binding and splicing activity. The failure of RNA interaction is proposed to lead to a phase-separation of the protein – directed by the C-terminal - and formation of insoluble aggregates that have characteristic similar to the pathological ones in terms of the phosphorylation and ubiquitination of the protein. Morato et al. makes use of an elegant amber suppression to introduce “authentic” acetyl-lisine in the protein sequence and further confirm the observations obtained with site-directed mutagenesis. Finally, they develop new specific antibodies and identify deacetylation of K136 by SIRT1. Overall, these results are fascinating and paint a picture where the strong regulation of post-translational modifications alter the localization and interaction of TDP-43 with other components. In this respect, the manuscript of Morato et al follows similar moves than the ones proposed by Cohen et al. (ref. 17) and provides corroboration that acetylations act as a switch for function and aggregation. The investigation of PTMs is fundamental for understanding the interplay between TDP-43, phase-separation, and neurotoxicity.

We are grateful for the reviewer's enthusiastic assessment.

1. The authors do not provide a strong motivation for the investigation of residue acetylation and its possible role in modulating cellular localization.

We have now updated the text referred by the reviewer. We refer now to the increasing interest in TDP-43 acetylation initiated by Cohen et al. We also refer now to the competition between ubiquitination and acetylation for lysine residues and to the well established role of ubiquitination in disease.

2. At line 66. the authors state that “we discovered that acetylation of K84 within the nuclear localization signal reduced nuclear import whereas acetylation of K136 in the RNA recognition domain impaired TDP-43 binding and splicing capabilities”. However, it was previously reported in literature that alterations of K84 can affect nuclear localization (e.g. Jiang et al, Scientific Report, 2017 use alterations in lysine and arginine residues within the NLS to suppress nuclear localization) and different localization and phase-separation of K136 mutants have been shown in ref. 16 (but not really discussed in the manuscript). A better contextualization of previous experiments and a critical indication on how specific mutations may affect the protein can help clarifying the original findings in the manuscript. One important advancement remains the testing of interactions of the protein with nucleic acids.

We agree that K84 is a well-established NLS residue, as discussed in the manuscript. The new finding here is its potential modulation by acetylation. We are not sure how relevant the study of Jiang et al. is to the present report, as it mostly deals with the N-terminal 1-77. While our study was in progress, Maurel et al. discovered K136 as a SUMOylation site. Consistent with our findings, they reported enhanced nuclear inclusion formation of a GFP-[K136R]TDP-43 fusion protein. However, based on structural considerations (new Fig. S2b), we believe that the K136R substitution causes artifactual perturbations of the RRM1. Here we show that it is the acetylation of TDP-43 K136 that can act as a potential physiological PTM reducing RNA binding and thus causing phase separation and TDP-43 pathogenesis. Circumventing problems of conventional site-directed mutagenesis, introduction of authentic acK136 by amber suppression confirmed the validity of this PTM.

3. The authors often use the term “aggregates” and the phase-separation terminology across the text is somehow confusing, since a phase separated granule is not an aggregate (perhaps it is an aggregation of multiple components, but the specific interaction at play usually confer a certain degree of fluidity, very specific concentration boundaries (binodal), and reversibility). The aggregate term usually invokes more static and possibly unspecific and/or irreversible formations. The manuscript may benefit of a better usage of nomenclature in this respect.

We have now restricted the use of the term “aggregates” and “phase separation” throughout the manuscript. Those terms are now used only when the phenomena observed can be described by them.

4. The nuclear “aggregates” of TDP-43 K136R mutant have a different distribution and size that does reported in ref. 16, where large “droplets” are observed. Can the author comment on this difference? Technically speaking, this is an important element in judging whether the protein is undergoing phase-separation or forming oligomers.

In our hands, [K136R]TDP-43 formed predominantly nuclear inclusions. The size of these inclusions was influenced by the time between transfection and fixation. In that regard, [K136R]TDP-43 inclusions behaved like [K136Q]TDP-43 (now quantified in Fig. 5e). Maurel et al. showed rather big inclusions in their study using GFP fusion proteins. Unfortunately, they did not specify the time passed after transfection for their immunofluorescence so we cannot be sure if this is indeed the differentiating factor. It would also be interesting to see a classification of aggregates to clarify how well those images represent the whole cell population. We did observe such big aggregates after 24 hours, but they were not the predominant size. In addition, Maurel et al. used different cell models (murine NSC34, motor neurons and HEK293T), which could display an accelerated phenotype compared to our HEK293E cell line. In our supplementary videos the change of size over time can be observed.

5. On a similar line, the authors report in Fig. 4 FRAP experiments from objects that are larger than 1 um and objects that are smaller than 0.5 um. A more detailed description of the experiments and choice of ROI would help the reader in contextualizing the experiments. Can the different velocity and amplitude of recover be related to the size of the objects? Or is there any clear evidence supporting that small “puncta” have physical properties different from “larger” ones (e.g. ones fusing, the other ones not)? The authors mention also “droplets fusing” at line 214-215. Can the authors quantify whether the average size of droplets is increasing over time (another quality of phase-separating systems) and whether there is a size threshold or time threshold that limit the growth? Have the authors tested FRAP across different delay times after droplets are formed to see whether recovery is time dependent (which would follow the hypothesis described in lines 216-217)?

As the reviewer suggests, we are proposing that smaller objects have a faster recovery rate than bigger aggregates. We have now added a comparison of mobile fraction at of the three groups (new Fig. 5c) where the differences can be seen more clearly. We have now quantified the size of droplets 24, 48 and 72 hours after transfection and the results suggest that the percentage of big aggregates in cells increase in a time-dependent manner (new Fig. 5e).

6. Have the authors tested the effect of temperature to verify if there is any transition in the solubilization of these puncta as a proxy for confirming the phase-separation nature and reversibility of their assembly vs. aggregation of the protein?

Certainly it would be interesting to see the effect of temperature on TDP-43 granules. However, heat shock has been reported to be a trigger of TDP-43 hyperubiquitination and mislocalisation into stress granules (Colombrita et al., 2009; Hans et al., 2020; McDonald et al., 2011). To validate the phase separation, we looked into the methodological guidelines established by Alberti et al. in their review “Considerations and challenges in studying liquid-liquid phase separation and biomolecular condensates”. From the proposed experiments, the only one that we could not perform was a concentration-dependent condensation assay due to the logarithmic nature of our overexpression system. In addition, we tried to establish a protocol using the phase separation inhibitor 1,6-hexanediol but the side effects of this treatment in cells precluded quantitative statements.

Reviewer #4

Morato et al., describes that acetylation at sites of Lysine 84 and Lysine 136 of TDP-43 regulates its cytoplasmic mislocalization and the aggregation propensity. Deacetylase SIRT1 is responsible for de-acetylation of TDP-43 at Lysine 136 and reduces the aggregation propensity of TDP-43. The strengths of manuscript include successfully incorporating authentic acetyl-lysine at the apparently regulatory sites in TDP-43 with amber suppression methodology. Overall, the data provided in this manuscript are convincing and with proper controls.

We thank the reviewer for the kind comments.

1. In Figure 1D, the bands for TDP-43 should be quantified by densitometry.

We have now added a quantification of the changes in the cytoplasmic fraction of Fig. 1d, normalized to wtTDP-43, in a new Fig. 1e.

2. In Figure 2D, both mutants, K136R and K136Q, increase ubiquitination and phosphorylation of TDP-43. The wild-type TDP-43 and authentic acetylation of K136 of TDP-43 should be compared for their ubiquitination and phosphorylation status to verify the crosstalk among acetylation, ubiquitination, and phosphorylation of TDP-43. The resulting data should be discussed in the discussion section.

We did look at the ubiquitination and phosphorylation of [acK136]TDP-43 but we could not see any detectable signal, both in western blots and immunofluorescence. We have now added a paragraph to the discussion commenting on this issue: “We think there are three factors that can influence this result. First, compared with transfected mutant [K136Q]TDP-43, expression of amber suppressed TDP-43 is much lower (see Fig. 6a), potentially below detection limit of other PTMs. The lower concentration might also not be sufficient to seed protein aggregation, phosphorylation and ubiquitination. Second, K136-acetyl in the amber suppressed TDP-43 can be removed by endogenous deacetylases such as SIRT1. This would attenuate any downstream effects of this acetylation. And third, as shown by Wang et al. (2018) glutamines favour the hardening of phase-separated RNA-binding molecules. While these key differences could be behind the lack of the strongest markers of TDP-43, amber suppression still could help identifying K136 as a residue crucial for RNA splicing and LLPS in TDP-43.”

While we have these concerns very present, we also believe that amber suppression allowed us to establish acetylation at K136 as a modulator of LLPS and RNA splicing.

REVIEWER COMMENTS

Reviewer #1 (Remarks to the Author):

Figure 1 shows data used to support the authors' conclusion that changes at residues K84 alter TDP-43 nuclear import. The levels of TDP-43 expression are more homogeneous than that presented in the original MS. However, the data remains weak depending on the methods used for the analysis. A "more representative image" is included in Fig 1C, showing cytoplasmic distribution of K84Q. Quantification shows 100% diffuse localization of the mutant in the cytoplasm. However, almost all the K84Q protein is found in the nucleus after biochemical fractionation in Fig 1D. Cytoplasmic localization of mutants including NLS is extremely low in Fig 1D, even if these levels are greater than WT. Likewise, the nuclear localization sequence (NLS) mutant is not included in Fig 1C, but is mostly cytoplasmic in the immunofluorescence images of Fig 4E. Again, the NLS mutant is almost all nuclear in Fig 1D. The fact that the authors observe similar results with K84Q and the NLS mutant of mostly nuclear presence may be due a problem with the fractionation assays. Similar nuclear/cytoplasmic fractionation of TDP-43 including a similar NLS mutation has been previously performed multiple times by others, showing much higher levels of TDP-43-NLS in the cytosolic fraction (e.g., Barmada S, 2010, J Neurosci). This discrepancy in the two different assays diminishes the strength of their model on the function of K84 modifications.

The new results to examine changes in binding affinity between WT and K136Q (Fig 3 B, C) to address point 4 of this reviewer are not acceptable. These assays and their analyses do not measure binding affinity, as stated in Pg. 6 of the Results. One of the problems is that the behavior of RNA-bound protein (nitrocellulose) and subsequent quantification highly suggest non-specific binding behavior. In addition, data in the plot is incompatible with the strong binding of (UG)₁₂ to WT TDP-43 previously established to be in the low nanomolar range. No affinity constant is derived from the plot and the figure and experiments are poorly described.

The new splicing experiments include all the mutants investigated, however, the results shown in Fig 3E are difficult to interpret because the PCR products for exon 9 inclusion are very hard to visualize. The authors should consider including a different more representative result to support the quantification of splicing regulation in Fig 3F.

Reviewer #2 (Remarks to the Author):

In this revised version the authors have added several critical new experiments, as well as clarified and corrected many obscure, confusing and erroneous statements and conclusions. I can therefore recommend this manuscript for publication.

Reviewer #3 (Remarks to the Author):

The revised manuscript addresses all my concerns and the results on the effect of acetylation on the cellular localization and phase-separation propensity of TDP-43 appears to be sound.

Summary from first round of review:

The authors report that K84 acetylation reduces the nuclear import, whereas K136 acetylation impaired RNA binding and splicing activity. The failure of RNA interaction is proposed to lead to a phase-separation of the protein – directed by the C-terminal - and formation of insoluble aggregates that have characteristic similar to the pathological ones in terms of the phosphorylation and ubiquitination of the

protein. Morato et al. makes use of an elegant amber suppression to introduce “authentic” acetyl-lisine in the protein sequence and further confirm the observations obtained with site-directed mutagenesis. Finally, they develop new specific antibodies and identify deacetylation of K136 by SIRT1. Overall, these results are fascinating and paint a picture where the strong regulation of post-translational modifications alter the localization and interaction of TDP-43 with other components. In this respect, the manuscript of Morato et al follows similar moves than the ones proposed by Cohen et al. (ref. 17) and provides corroboration that acetylations act as a switch for function and aggregation. The investigation of PTMs is fundamental for understanding the interplay between TDP-43, phase-separation, and neurotoxicity.

Reviewer #1

Figure 1 shows data used to support the authors' conclusion that changes at residues K84 alter TDP-43 nuclear import. The levels of TDP-43 expression are more homogeneous than that presented in the original MS. However, the data remains weak depending on the methods used for the analysis. A "more representative image" is included in Fig 1C, showing cytoplasmic distribution of K84Q. Quantification shows 100% diffuse localization of the mutant in the cytoplasm.

We are sorry for our misleading classification terms. In Fig. 1c the orange bars represent percentage of cells with "nuclear + cytosolic, diffuse" TDP-43 and yellow cells with "nuclear + cytosolic, clumpy". This was clarified in the figure label and legend. Moreover, the images in Fig. 1b were enlarged for better visibility.

However, almost all the K84Q protein is found in the nucleus after biochemical fractionation in Fig 1D. Cytoplasmic localization of mutants including NLS is extremely low in Fig 1D, even if these levels are greater than WT. Likewise, the nuclear localization sequence (NLS) mutant is not included in Fig 1C, but is mostly cytoplasmic in the immunofluorescence images of Fig 4E. Again, the NLS mutant is almost all nuclear in Fig 1D. The fact that the authors observe similar results with K84Q and the NLS mutant of mostly nuclear presence may be due a problem with the fractionation assays. Similar nuclear/cytoplasmic fractionation of TDP-43 including a similar NLS mutation has been previously performed multiple times by others, showing much higher levels of TDP-43-NLS in the cytosolic fraction (e.g., Barmada S, 2010, J Neurosci). This discrepancy in the two different assays diminishes the strength of their model on the function of K84 modifications.

We are irritated ourselves by the apparent discrepancy of nucleocytoplasmic distribution detected by immunofluorescence staining and biochemical fractionation. Although immunostaining might introduce some technical artefacts due to fixation, cell permeabilization, organelle preservation, antibody penetration and epitope accessibility, we tend to rely on this method for the somewhat expected conclusion that modification of the well-known residue K84 impairs nuclear import of TDP-43. The biochemical fractionation protocols involve more sample processing steps, some of which might affect the preservation of TDP-43 localization. To avoid distraction from the main message, we have moved the biochemical fractionation experiments to the supplement (Fig. S1b,c and new d).

We tried alternative fractionation protocols, and show the results (new Fig. S1d) from a more comprehensive fractionation assay (Subcellular Protein Fractionation Kit for Cultured Cells, #78840, Thermo Fisher). Again supporting the major point, [K84Q]TDP-43 (and [Δ NLS]TDP-43) band strengths were stronger than those of wt and the conservative mutant [K84R]TDP-43. Because relatively more total cytosolic protein was loaded in Fig. S1d compared to Fig. S1b, cytosolic TDP-43 "leakage" upon fractionation appears more prominent. There is some cytosolic fraction of the small histone protein H3, but soluble nuclear proteins such as YY1 and importantly hnRNPA1 that should behave similar to TDP-43 in terms of nuclear localization do not deviate as much from the expected partitioning as does TDP-43. As the same discrepancy between immunostaining and biochemical fractionation is seen in the mentioned paper (Barmada et al. 2010 compare Fig. 7D and Fig. 6A), we have to assume for the time being that biochemical fractionation is not the most reliable method to determine the exact nucleo:cytoplasmic ratio of TDP-43. Differences in

nuclear and cytosolic TDP-43 solubility may introduce another confounding factor. Thus, the qualitatively supportive biochemical fractionation data showing relatively more cytoplasmic mislocalized [K84Q]TDP-43 (and [Δ NLS]TDP-43) is now moved into the supplementary information.

The new results to examine changes in binding affinity between WT and K136Q (Fig 3 B, C) to address point 4 of this reviewer are not acceptable. These assays and their analyses do not measure binding affinity, as stated in Pg. 6 of the Results. One of the problems is that the behavior of RNA-bound protein (nitrocellulose) and subsequent quantification highly suggest non-specific binding behavior.

In addition, data in the plot is incompatible with the strong binding of (UG)12 to WT TDP-43 previously established to be in the low nanomolar range. No affinity constant is derived from the plot and the figure and experiments are poorly described.

The binding affinities referred to (e.g. Buratti and Baralle. 2001, Bhardwaj et al. 2013 or Kitamura et al. 2018) were determined with purified recombinant TDP-43, mostly the isolated RRM domain. Here we attempt something beyond, namely the assessment of RNA binding to native full-length TDP-43 pulled down from eukaryotic cells. In the revised Fig. 3b,c, we show extended concentration ranges demonstrating a saturation curve rather than non-specific binding. However, at higher protein input we run into overload problems with the dot blot assay, preventing perfect K_D determinations particularly for the binding-deficient mutant TDP-43. We acknowledge this assay limitation seen in revised Fig. 3b in the figure legend. The association constant can only be roughly estimated using this novel assay. We revised the results text using descriptive terms (“binding strength”) rather than “affinity” or alike. The dot-blot experiment presented should be taken as a qualitative proof of the different binding properties of wt and K136Q TDP-43. Together with the results of pull-down (Fig. 3a) and the splicing activity reduction (Fig. 3e,f) as well as the structure modelling (Fig. S2), we believe our data indicates the reduced RNA interaction of TDP-43 when acetylated at K136.

The new splicing experiments include all the mutants investigated, however, the results shown in Fig 3E are difficult to interpret because the PCR products for exon 9 inclusion are very hard to visualize. The authors should consider including a different more representative result to support the quantification of splicing regulation in Fig 3F.

We have now added a brighter image of the CFTR's products, paying special attention to the visibility of the exon 9 inclusion bands.

Reviewer #2

In this revised version the authors have added several critical new experiments, as well as clarified and corrected many obscure, confusing and erroneous statements and conclusions. I can therefore recommend this manuscript for publication.

Thank you.

Reviewer #3

The revised manuscript addresses all my concerns and the results on the effect of acetylation on the cellular localization and phase-separation propensity of TDP-43 appears to be sound.

We thank the reviewer for the good reception of the revised manuscript.

REVIEWERS' COMMENTS

Reviewer #1 (Remarks to the Author):

The reviewed manuscript addresses this reviewer's major concerns.

Reviewer #1

The reviewed manuscript addresses this reviewer's major concerns.

We thank the referee's insightful comments throughout the review, which helped us to continuously improve our manuscript. We are happy to have settled the major concerns.